# Increased risk of near term global warming due to a recent AMOC weakening

Rémy Bonnet[1✉], Didier Swingedouw[2], Guillaume Gastineau[3], Olivier Boucher[1], Julie Deshayes[3], Frédéric Hourdin[4], Juliette Mignot[3], Jérôme Servonnat[5] & Adriana Sima[4]

Some of the new generation CMIP6 models are characterised by a strong temperature increase in response to increasing greenhouse gases concentration[1]. At first glance, these models seem less consistent with the temperature warming observed over the last decades. Here, we investigate this issue through the prism of low-frequency internal variability by comparing with observations an ensemble of 32 historical simulations performed with the IPSL-CM6A-LR model, characterized by a rather large climate sensitivity. We show that members with the smallest rates of global warming over the past 6-7 decades are also those with a large internally-driven weakening of the Atlantic Meridional Overturning Circulation (AMOC). This subset of members also matches several AMOC observational fingerprints, which are in line with such a weakening. This suggests that internal variability from the Atlantic Ocean may have dampened the magnitude of global warming over the historical era. Taking into account this AMOC weakening over the past decades means that it will be harder to avoid crossing the 2 °C warming threshold.

[1] Institut Pierre-Simon Laplace, Sorbonne Université/CNRS, Paris, France. [2] Environnements et Paléoenvironnements Océaniques et Continentaux, Université de Bordeaux/CNRS, Bordeaux, France. [3] Laboratoire d'Océanographie et du Climat: Expérimentations et Approches Numériques, Institut Pierre-Simon Laplace, Sorbonne Université/CNRS/IRD/MNHN, Paris, France. [4] Laboratoire de Météorologie Dynamique, Institut Pierre-Simon Laplace, Sorbonne Université/CNRS/Ecole Normale Supérieure/Ecole Polytechnique, Paris, France. [5] Laboratoire des Sciences du Climat et de l'Environnement, Institut Pierre-Simon Laplace, CEA/CNRS/UVSQ, Gif-sur-Yvette, France. ✉email: remy.bonnet@ipsl.fr

Projections of future climate from Earth System Models (ESM) represent a crucial source of information for adaptation and mitigation planning. The Equilibrium Climate Sensitivity (ECS), defined as the warming resulting from a $CO_2$ doubling once the system has equilibrated, as well as the Transient Climate Response (TCR), defined as the warming produced after 70 years by a 1% per year increase in $CO_2$ up to a doubling of $CO_2$, are often used to estimate the sensitivity of ESMs to anthropogenic $CO_2$ emissions. Some models from the Coupled Model Intercomparison Project phase 6 (CMIP6) exhibit relatively large climate sensitivities. Indeed, one-third of the models[1] have an effective ECS greater than the IPCC AR5 "likely" range (1.5–4.5 K) and one-fifth of the models[2] have a TCR greater than the IPCC AR5 "likely" range (1–2.5 K). Consistently, these models with higher sensitivity project greater warming over the 21st century than previously reported in CMIP5, although a direct comparison is challenging because emission scenarios used to drive the models have also evolved. These larger sensitivities from some of the CMIP6 models are also difficult to reconcile with estimates of ECS and TCR based on observations over the historical period[3–5] (renamed S_hist and TCR_hist hereafter), which are around 2.5 K for S_hist according to a recent study[6] and around 1.6 K for TCR_hist[5]. The associated uncertainties to S_hist and TCR_hist, however, are large. Alternative estimates using the observed variability have also been proposed[7], but do not allow us to rule out high ECS values.

This new generation of models with large effective ECS raises important questions: (i) are these models realistic in comparison to the last few decades of climate observations? (ii) how to interpret these highly sensitive models in relation to climate change over the historical era? If these models are not falsifiable, it would imply higher risks and costs induced by future climate change impacts and the need for greater and faster mitigation efforts to achieve climate targets than previously thought. To address these issues, recent studies[2,8,9] tried to constrain CMIP6 climate model projections with observed warming trends over the last decades. Although it makes sense to focus on the last few decades to estimate the impact of greenhouse gases (GHGs), because aerosol forcing has not changed much, there is a need to understand the model spread and to explore and attribute the recent observed changes. In order to take into account all the information available over the historical period, a recent study[10] developed a new statistical method, leading to a reduction of the uncertainty on the projected future warming by about 50% and a slightly higher future warming in CMIP6 relative to CMIP5. However, the authors showed that this method poorly accounted for the internal variability in some CMIP6 models[10] that exhibit stronger multi-decadal to centennial internal climate variability than CMIP5 models[11], which might have strong implications. Indeed, the low-frequency internal variability at decadal to multi-centennial time scales can temporarily enhance or reduce the long-term imprints of externally forced climate change. In particular, such variability might have a strong impact on the way climate models should be compared to observations.

In this study, we address these issues highlighting new perspectives, which allows us notably to evaluate the risk of larger warming over the next decades, using the IPSL-CM6A-LR model. This model is one of those new generation models characterized by a rather large sensitivity, close to the upper bound of the constrained projections, with an effective ECS of 4.5 K—computed from a 4xCO₂ abrupt forcing experiment and a 150-year regression—and a TCR of 2.4 K. It is also characterized by a relatively high low-frequency internal climate variability in comparison to the CMIP6 models[11]. Taking advantage of CMIP6, the Institut Pierre-Simon Laplace Climate Modelling Centre has produced an ensemble of extended historical simulations[12] (referred to as IPSL-EHS, see Methods). By using this ensemble, we show the influence of multi-centennial internal climate variability on the Global near-Surface Air Temperature (GSAT) warming since the middle of the 20th century, as well as the related uncertainties on the estimation of S_hist and TCR_hist. The spread in the GSAT trends is related to trends in AMOC. Specifically, we show that members matching best the observed record of GSAT are also those with a large internally-driven weakening of the AMOC in the past decades. Several AMOC fingerprints confirm the consistency of the AMOC variations in these members with observations. The mechanisms involved are also confirmed by the analysis of other climate model ensembles. Over the next decades, the low-frequency internal variability in these members leads to slightly more warming than the rest of the ensemble.

## Results and discussion

**Climate sensitivity of IPSL-CM6A-LR over the historical period.** The IPSL-CM6A-LR ensemble mean shows a larger than observed warming over the historical period (Fig. 1a). This global warming bias can be due to (i) an overestimated climate sensitivity in the model, (ii) an overestimation of the net anthropogenic radiative forcing, or (iii) multi-decadal to multi-centennial internal variability which offsets some of the warming in the observations. While the radiative forcing by aerosols in IPSL-CM6A-LR is relatively weak at $-0.6$ $Wm^{-2}$, which might be in favor of hypothesis (ii), we argue that this is compensated by an underestimation of the (positive) radiative forcing by non-$CO_2$ greenhouse gases[13]. Thus, although we cannot discard hypothesis ii) in the light of the large uncertainties on the aerosol forcing, we focus this study on the exploration of hypotheses (i) and (iii).

Despite the warmer than the observed ensemble mean, some members have a consistent representation of the historical and more recent warming trends (Fig. 1a and Supplementary Fig. 3a). Member #14 for example, has one of the best representations of the GSAT in comparison to observations, with the lowest root mean square error (RMSE) of 0.14 K for the annual GSAT anomaly over the 1900–2018 period (Supplementary Fig. 3b).

We estimate S_hist and TCR_hist for each individual member of the IPSL-EHS using a similar method as former studies[3,4] based on the changes of surface air temperature in 1999–2018 compared to the preindustrial period (see Methods, Fig. 1b, c). The ECS value of 4.5 K diagnosed from the abrupt-4xCO₂ experiment is at the top of the S_hist range inferred in the IPSL-EHS, which is consistent with the previous studies[6,14]. This is likely due to the non-linear relationship between radiative forcing and the change in GSAT as a result of stronger feedbacks in a warmer climate[15]. The pattern effect might also contribute. Indeed, the short-term warming of models has a pattern of surface air temperature warming different from the stabilized warming pattern leading to weaker low cloud feedback. The range of S_hist values is indeed more consistent with the value diagnosed from an abrupt-2xCO₂ experiment at 3.8 K. The same is true for the TCR value of 2.4 K diagnosed from the 1pctCO₂ experiment which is larger than the mean of TCR_hist values. More importantly, the spread in modelled S_hist and TCR_hist values is substantial ($\pm 2\sigma = 1.25$ K), suggesting such values are largely influenced by internal climate variability (Fig. 1b, c). If the low-frequency internal variability simulated by the model is realistic then we should expect a large uncertainty in observational estimates of S_hist, which question its definition and its utility for assessing correctly future greenhouse emissions admissible to remain below-given thresholds in GSAT. Furthermore, this low-frequency internal variability presumably strongly interacts with the transient response to external forcing and thereby affects TCR_hist.

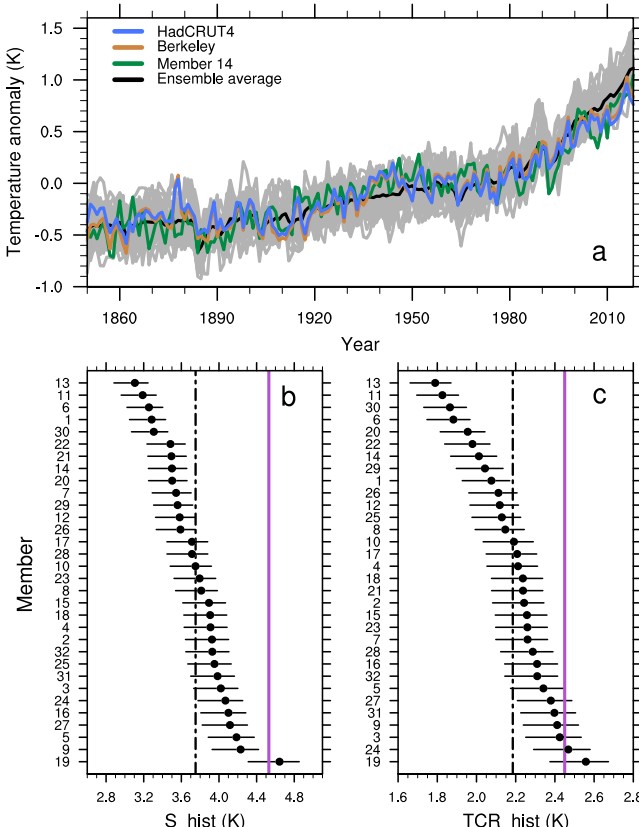

**Fig. 1 Sensitivity of IPSL-CM6A-LR over the historical period. a** Global mean near-surface air temperature (GSAT) anomaly (K) relative to the 1880–2018 period for the ensemble average (black), individual members (gray), and member #14 (green) of the IPSL ensemble of extended historical simulations (IPSL-EHS), the infilled HadCRUT4-CW[41,42] (blue) and the Berkeley[45] observational datasets (brown). **b** The climate sensitivity calculated over the historical period (S_hist, in K) for each individual member of the IPSL-EHS ranked from the lowest to the highest value (see Methods). The black dot-dashed line indicates the ensemble mean of S_hist values and the purple line indicates the Equilibrium Climate Sensitivity (ECS) value[13] of the IPSL-CM6A-LR model. **c** Same as **b** but for the transient climate response calculated over the historical period (TCR_hist, in K), the purple line indicating the Transient Climate Response (TCR) value[13] of IPSL-CM6A-LR. Error bars in **b** and **c** are calculated using the lowest and largest values of the forcing due to a doubling in atmospheric $CO_2$ concentration (see Methods for more details).

**Low-frequency internal climate variability of the global surface temperature**. IPSL-CM6A-LR is characterized by a multi-centennial variability of AMOC and GSAT in the piControl simulation[13]. A recent study[16] on the associated mechanisms suggests that a positive AMOC anomaly is associated with warm temperature anomalies in the Northern Hemisphere, a reduced Arctic sea-ice extent, and salinity-driven positive density anomalies in the Nordic Seas and in the North Atlantic subpolar gyre. During a positive AMOC, both ocean and sea ice freshwater export from the Arctic Ocean decreases, which contributes to build a large freshwater anomaly in the top 100 m of the Arctic Ocean, while the Atlantic inflow of salty water into the Arctic increases the salinity below 100 m. After several decades, the freshwater anomaly covers most of the western Arctic which eventually increases the oceanic freshwater export from the Arctic into the Nordic Seas through the Fram Strait. This slows down the AMOC. The Northern Hemisphere climate then cools and Arctic sea ice grows, reversing the phase of the cycle. The 32

members of the ensemble were run with the same external forcings but initialised from different years in the pre-industrial control simulation in order to sample this multi-centennial variability. Due to the link between the AMOC variability and the initial state used in the historical simulations (Supplementary Fig. 4), a clear positive significant relationship exists between the AMOC trend in 1850-2010 and the S_hist and TCR_hist.

A recent AMOC reconstruction[17], using an index (named "Caesar index" hereafter) based on the relation between the North Atlantic Subpolar Gyre Sea Surface Temperature (SST) relative to the global SST from November to May and the AMOC, suggests a weakening of the AMOC since the 1940s. Although some debate exists concerning the robustness of this index, the implied AMOC weakening trend is supported by other reconstructions and indices[18]. There are still substantial uncertainties though, and other AMOC fingerprints will be considered later on in this study.

A significant positive relationship is found in IPSL-EHS between GSAT and AMOC trends over the 1940–2016 period (when AMOC weakening is suggested to start in the reconstruction), with a determination coefficient $r^2=0.82$ (Fig. 2a), in line with previous studies[19–21]. Unlike the historical period, no clear relationship is found between the AMOC trends over the 1940–2016 period and the calculated values of S_hist or TCR_hist, as these quantities rely on the temperature difference between the pre-industrial (1850–1879) and present-day (1999–2018) periods. Figure 2a shows that ensemble members with the weakest GSAT warming are those with the strongest AMOC weakening and the members with the largest GSAT warming are those with the strongest AMOC strengthening. The members whose warming levels match best the GSAT observations both in terms of the recent trends and RMSE over the 1900–2018 period are also characterized by a negative AMOC trend (Fig. 2a, Supplementary Fig. 3) suggesting a possible link. Member #14 of IPSL_EHS in particular is again one of the closest members to observations in terms of both the trends in reconstructed AMOC index[17] and GSAT. We pre-select members that are consistent with the observed GSAT and AMOC trends, and then select the six members with the lowest GSAT RMSE among those. The subset is composed of members #14, #4, #5, #25, #29, and #30, with RMSE ranked 1st, 8th, 2nd, 3rd and 5th, and 7th, respectively.

These members are characterized by a strengthening of the AMOC until the 1940s followed by a decline, although member #30 shows a different evolution from the other members until the 1920s (Fig. 2b). Conversely, the forced AMOC estimated by the ensemble mean shows only weak changes, with a small strengthening up to the 1990s followed by a weakening, as found in other CMIP6 models[22]. The evaluation of the relative influence of internal variability and forced response in the AMOC weakening of these members indicates that internal climate variability has been the main driver of the AMOC variability in IPSL-EHS since the 1940s (Supplementary Section 1). From the 21st century onwards, however, external forcings have an increasing influence on internal variability, and dominate the decline in AMOC (Fig. 2b). Consequently, the AMOC weakening suggested in the Caesar reconstruction[17] might be mainly internally-driven rather than externally forced. Conversely, an internally generated enhancement of the AMOC in the early 20th century might have warmed GSAT at that time, in agreement with other studies[23], suggesting an important role of internal variability for this early century warming. This is consistent with the increase of the Caesar index over the beginning of the 20th century (Fig. 3a).

We now check whether the members identified previously present realistic trends in North Atlantic SST or are characterized by excessive cooling in comparison to the observations and the

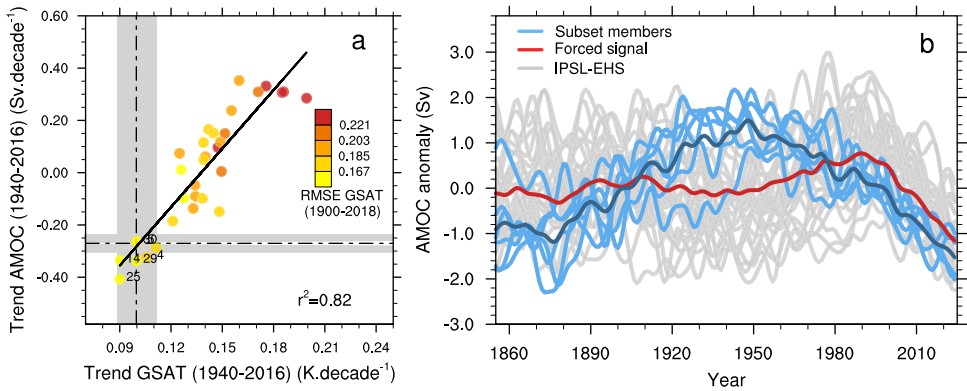

**Fig. 2 Relationship between GSAT and AMOC over the historical period. a** Scatter plot of Global near-Surface Air Temperature (GSAT) (K decade$^{-1}$) versus Atlantic Meridional Overturning Circulation (AMOC) (Sv decade$^{-1}$) trends calculated over the 1940-2016 period from the IPSL ensemble of extended historical simulations (IPSL-EHS; filled circles) and the observations (dot-dashed lines), with HadCRUT4-CW[41,42] for the temperature and the Caesar index[17] as a proxy for the AMOC evolution, with the related uncertainties (gray). The black solid line represents the least square regression between these two variables in IPSL-EHS, with a determination coefficient $r^2 = 0.82$ ($p < 0.1$, see Method). The color scale represents the RMSE between simulated and observed (HadCRUT4-CW[41,42]) annual GSAT anomalies over the period 1900–2018. **b** Time evolution of the low-pass filtered AMOC strength anomaly from IPSL-EHS (gray), the AMOC forced signal (or ensemble mean, red), and the subset of 6 members labelled in **a** (light blue), with the subset average (dark blue). The anomaly is calculated for each member with respect to its 1850–2018 average. A Lanczos low-pass filter with a cutoff period of 11 years is used.

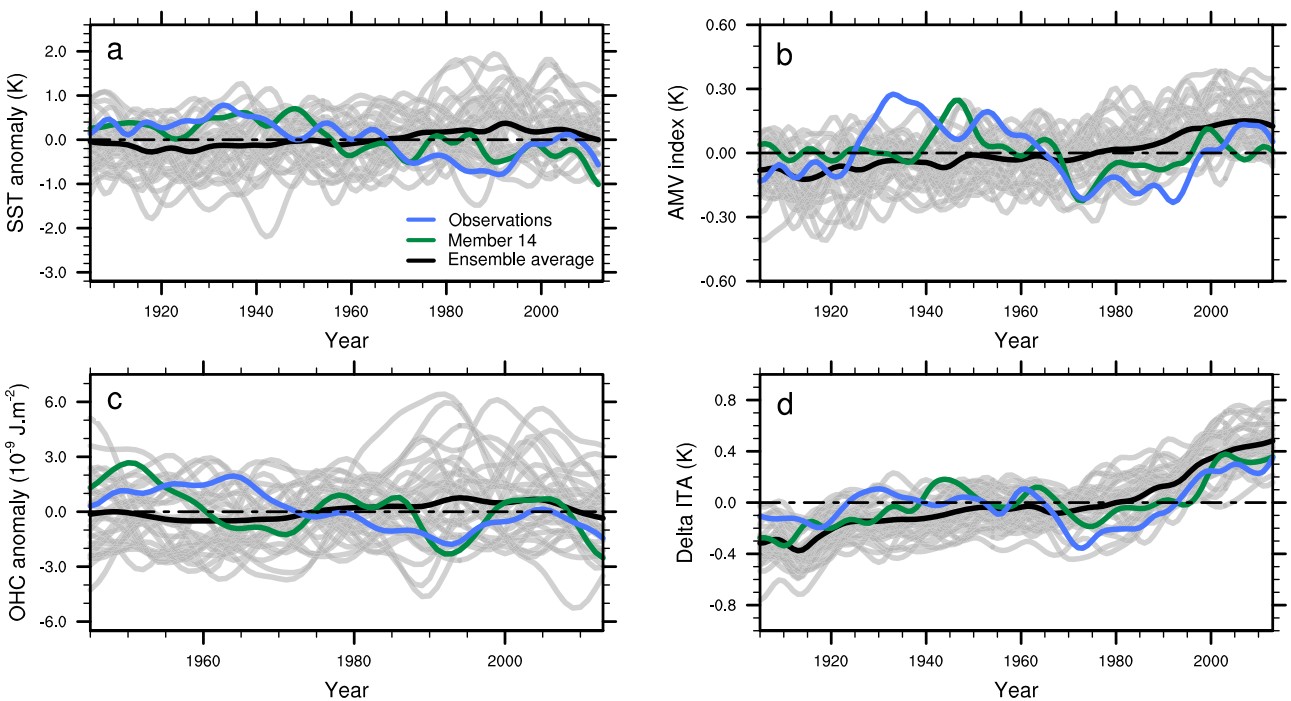

**Fig. 3 Evaluation of Atlantic Meridional Overturning Circulation fingerprints. a** Time evolution of the Caesar index computed as the low-pass filtered sea surface temperature anomaly (K) averaged over the North Atlantic Subpolar Gyre from November to May from the ERSSTv5 observational dataset[54] (blue), the 32 historical members (gray) with the ensemble mean (black) and member #14 (green). **b**, Same as **a** but for the Atlantic Multidecadal Variability index (see Methods). **c** Same as **a** but for the Ocean Heat Content (OHC) anomaly (10$^9$ J m$^{-2}$) between 0 and 700 m averaged over the Newfoundland region minus the OHC averaged over the North Atlantic Subpolar Gyre region from the IAP observational dataset[55,56] (blue) **d** Same as **a** but for the interhemispheric near-surface air temperature difference (K), with the HadCRUT4-CW observational dataset[41,42] (blue). A Lanczos low-pass filter with a cutoff period of 11 years is used in **c** and **d** and the anomalies are computed over the displayed period.

rest of the ensemble (Supplementary Fig. 5). Over the 1940-2018 period, member #14 has an averaged SST trend of 0.06 K decade$^{-1}$ over the North Atlantic, slightly weaker than the observed trend of 0.065 K decade$^{-1}$ (Supplementary Fig. 5). The pattern is overall close to the observations, with a spatial correlation coefficient of 0.73. The subset of members previously identified are also among the members with average trends over the North Atlantic closest

to the observations. Due to its strong AMOC weakening (Fig. 2a), member #25 shows the smallest SST trend in the North Atlantic with a value of 0.042 K decade$^{-1}$.

**Further lines of evidence from AMOC fingerprints.** In order to strengthen the hypothesis that a weakening of the AMOC in the

real world since the middle of the 20th century may have masked a fraction of the anthropogenic global warming, we evaluate four different observable AMOC fingerprints against our ensemble members. A first SST fingerprint is calculated following the Caesar index[17]. Then, we use the Atlantic Multidecadal Variability[24–26] index (AMV index, see Methods), as basin-wide low-frequency variations in the North Atlantic SST are related to the AMOC low-frequency variability in many climate models[27]. Whether low-frequency variations in AMOC and AMV are the results of naturally-occurring internal variability, external forcings, or a mix of both is still debated[27–29]. This is also the case in the IPSL-EHS, with significant Pearson correlations ranging from 0.56–0.97 between the AMOC and the AMV index over the 1850-2018 period in the various members (see Methods). We calculate a third index based on the upper (0–700 m) Ocean Heat Content (OHC) averaged offshore Newfoundland minus the average over the North Atlantic Subpolar Gyre (ΔOHC). The upper OHC is indeed significantly correlated with the AMOC in IPSL-CM6A-LR over these two regions, with a negative correlation offshore Newfoundland and a positive correlation over the North Atlantic Subpolar Gyre (Supplementary Fig. 6a). This correlation dipole is consistent with observations and other climate model simulations[30]. As no significant correlations are found between the upper OHC and the AMOC forced signal, these correlations are mainly due to the AMOC internal variability (Supplementary Fig. 6b). Finally, the fourth index is defined as the difference between the Northern and the Southern Hemisphere anomalies of near-surface air temperature (ΔITA), which might also be a good indicator of the AMOC low-frequency variability, given its impact on interhemispheric temperature[31]. The time series of these four observation-based indices are included within the range of IPSL-EHS (Fig. 3), which gives confidence in the capacity of the model to represent climate dynamics accurately.

Furthermore, a good consistency is found between the six members identified above as best fitting GSAT over the observed period and the observations for the four AMOC fingerprints. In member #14 for example, Pearson correlations amount to 0.48 for the Caesar index (Fig. 3a), 0.49 for the AMV index (Fig. 3b), 0.46 for the ΔOHC index (Fig. 3c) and 0.74 for the ΔITA index (Fig. 3d) and are significant for the Caesar index, the ΔOHC index and ΔITA index (see Method). Notwithstanding, the decrease of $-0.13$ K decade$^{-1}$ of the Caesar index in member #14 since the 1940s, consistent with a weakening AMOC, is close to the observed trend of $-0.11$ K decade$^{-1}$. The decrease in the observed ΔOHC index of $-0.45 \times 10^9$ J m$^{-2}$ decade$^{-1}$ is also close to observations in member #14, with a decrease of about $-0.42 \times 10^9$ J m$^{-2}$ decade$^{-1}$. The pattern of the observed North Atlantic upper OHC trend calculated over the 1940-2020 period is consistent with a weakening AMOC, with significant positive trends over Newfoundland and significant negative trends over the Subpolar Gyre (Supplementary Fig. 6c). This pattern is no longer visible considering only the last decades (1980–2020), a period more influenced by the external forcing and characterized by strong warming of the upper OHC over the polar region (Supplementary Fig. 6d). The fact that these six members whose AMV is overall in phase with observations are those with a GSAT evolution most consistent with observations underscores the important role of the AMV in the GSAT variability at least in this model. This is consistent with a recent study[32] showing the main role of the AMV in the internal variability of GSAT on multi-decadal time scales. This good consistency is reinforced by the fact that member #14 has low RMSEs for these four AMOC fingerprints.

Member #14 is not the only one with overall low RMSEs for these four AMOC fingerprints. Members #29, #4, #5, #25, and #30, identified previously, also have overall the lowest RMSEs in

comparison to the rest of the ensemble (Supplementary Fig. 6). Notwithstanding, the consistency between AMOC fingerprints in the subset of members identified in Fig. 2a and in observations reinforces the hypothesis that the AMOC may have been weakening since the second half of the 20th century, thereby masking a fraction of the anthropogenic global warming as seen in the IPSL-EHS.

**Multi-model analysis.** We now examine how other climate models support our analysis of IPSL-EHS. A similar significant positive relationship between the AMOC and the GSAT 60-year trends is found in the piControl simulation of IPSL-CM6A-LR, as well as in ten out of thirteen other CMIP6 models (IPSL-CM6A-LR, both CNRM models, MPI-ESM1-2, CESM2-WACCM, SAM0-UNICON, CMCC-CM2, MRI-ESM2, EC-Earth3, and CIESM) (Fig. 4a). Those models also generally show significant multi-centennial AMOC variability compared to red noise at periods longer than 100 years (Fig. 4b). The models with the strongest AMOC-GSAT trends relationships are also those with the largest GSAT variance in their piControl simulations. Taken together these results highlight the role of internal variability in these models and suggest that the relationship between the AMOC and GSAT trends found in IPSL-CM6A-LR is rather widespread in CMIP6 models albeit with varying strength and range.

Only a few models with large historical ensembles are available to repeat the analysis performed with the IPSL-EHS: MPI-ESM1.1 with a low ECS, CNRM-CM6A-1, and CanESM5 with a high ECS. We investigate the consistency of our previous results with these models in Supplementary Section 2, which is summarized here. First, one can see that some members in the CNRM and the MPI ensembles reproduce the observed warming (similar GSAT trend as observations), unlike the CanESM5 ensemble. This latter model is therefore omitted. This may be due to its large climate sensitivity and/or its weak internal climate variability. Furthermore, a similar positive relationship between AMOC and GSAT trends is found in the CNRM historical simulations as in the IPSL-EHS, with, nevertheless, larger regression coefficients due to the larger internal climate variability of CNRM-CM6A-1[11]. A positive relationship is also found in the MPI ensemble, with a surprisingly large fraction of the historical members showing larger GSAT trends than observed despite a low climate sensitivity for this model. This might be due to its weak aerosol forcing. In addition, the members which fit best the GSAT and the AMOC weakening also match best the AMOC fingerprints in the CNRM-CM6 ensemble (Supplementary Fig. 9). This is not the case for the MPI ensemble where these members are scattered throughout the other members of the ensemble for the three fingerprints, which means that the members of the MPI ensemble with a weakening AMOC do not project well onto the observed AMOC fingerprints (Supplementary Fig. 8).

In conclusion, the analysis of three other large ensembles of historical simulations does not invalidate our results that suggest that the weakening in AMOC reconstructions since the second half of the 20th century might be due to internal variability. Indeed, similar behaviour can be seen in one model (CNRM-CM6 with a high ECS), the CanESM5 model does not reproduce the IPSL-CM6A-LR features as it appears to have warming not compatible with the observations, and the MPI-ESM1.1 model, in which the members with an AMOC trend consistent with a recent reconstruction[17] are not those with the best representation of the four observed AMOC fingerprints analyzed in this study.

**Implications for future global warming and discussion.** Our analysis suggests that a fraction of anthropogenic warming might

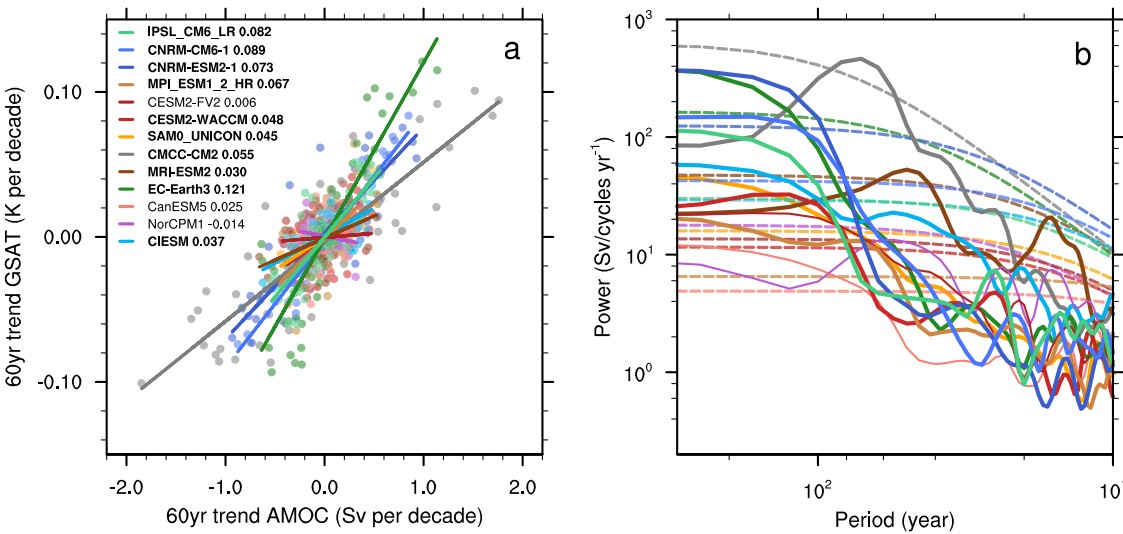

**Fig. 4 Relationship between GSAT and AMOC in CMIP6 control simulations. a** Scatter plot of Atlantic Meridional Overturning Circulation (AMOC) (Sv decade$^{-1}$) versus Global near-Surface Air Temperature (GSAT) (K decade$^{-1}$) trends calculated overall 60 year windows increasing by 10 years from the first 500 years of the pre-industrial control (piControl) simulations of thirteen CMIP6 models. The lines represent the least square regression between these two variables for each model. The regression coefficients (K Sv$^{-1}$) are indicated in bold beside the model's name when the regression is significant ($p < 0.1$, see Methods). **b** Smoothed power spectra of the AMOC time series from the first 500 years of the piControl simulations of the same thirteen CMIP6 models (solid lines, thick when the regression is significant), with the 95% confidence limit estimated from a red noise (dashed lines). A Lanczos low-pass filter with a cutoff period of 5 years is used.

have been hidden by an AMOC weakening that started in the middle of the 20th century and is mainly related to internal climate variability. Indeed, the subset of members identified previously, which is characterized by a strong AMOC weakening over the 1951–1990 period in comparison to the IPSL-EHS (Fig. 5a, b) shows a lower warming over the same period relative to the full IPSL-EHS (Fig. 5c, d). The same subset of members experiences an internally-generated AMOC strengthening over the next few decades. This is associated with a larger warming rate of about 0.36 K per decade relative to the IPSL-EHS mean of 0.34 K per decade. This influence of a phase change in the AMOC is however limited, as both the AMOC and GSAT low-frequency internal variability are projected to decrease in response to external forcings[33]. Taken together, these results reinforce the risk of crossing the 2 °C warming objective. Indeed, our subset of members with a low-frequency variability consistent with the observations shows larger warming than the ensemble means over the next decades. Moreover, taking into account this AMOC weakening over the historical era reinforces the risk of faster warming, as a fraction of anthropogenic global warming, which could have been hidden by this low-frequency internal variability, is expected to materialize in the coming decades. These results thus seem to be in line with a recent study[34] suggesting that the pattern effect could be in part related to internal climate variability. This internally-driven pattern effect could have masked part of human-induced global warming in recent decades. Nevertheless, the time period investigated here is longer than the last four decades analyzed in this other study, and our mechanism is related to the North Atlantic variability, rather than the Pacific variability, as found in the previous studies[35]. Future work is required to quantify how accounting for North Atlantic low-frequency variability, both in the Atlantic and Pacific oceans, modifies observational constraints on future warming levels.

The realism of the multi-centennial low-frequency internal variability found in some of the CMIP6 models[10,11] is a crucial element of our results. Although the instrumental period is short,

previous studies suggested that the Atlantic Multidecadal Variability is underestimated in CMIP5 models, with a persistence lower than the one deduced from observations[36,37]. To gain insight on this realism, paleodata provides additional information, but it usually suffers from large uncertainties concerning quantified estimations of the variability, especially at large spatial scales. While it is usually believed that model simulations might have too low multi-centennial variability as compared to proxy records[38], a recent study[11] suggests that GSAT interdecadal variability of some CMIP6 models might be overestimated over the period 1450–1840 in comparison to the pre-industrial control simulations. Therefore, we cannot exclude that some CMIP6 models, such as IPSL-CM6A-LR have too much internal variability. On the longer time scale of the last two thousand years, the different methods to reconstruct global mean surface temperature do show considerable uncertainty in terms of the magnitude of the multi-centennial variability[39], while the external forcings and their impact remain poorly estimated in models. Therefore, it seems difficult at the moment to properly assess the realism of model simulations with those reconstructions. Such an evaluation of model simulations might deserve a dedicated analysis using the PAGES2K database and last millennium simulations from CMIP6, using advanced techniques like pseudo-proxy approaches[40] in order to compare model simulations and reconstructions in a coherent framework.

To conclude, we have shown here that the different indications of an AMOC weakening since the mid-20th century might be mainly of internal origin coming from multi-centennial variability of the ocean. If true, this might mean that transient climate sensitivity estimated from the observational records, especially over the last 6–7 decades may be underestimated. Thus, emergent constraint approaches that try to constrain future warming using the recent decades should fully embrace the issue of low-frequency internal variability and take into account individual ensemble members rather than ensemble means, as this might have crucial implications in terms of how different models are weighted in such studies.

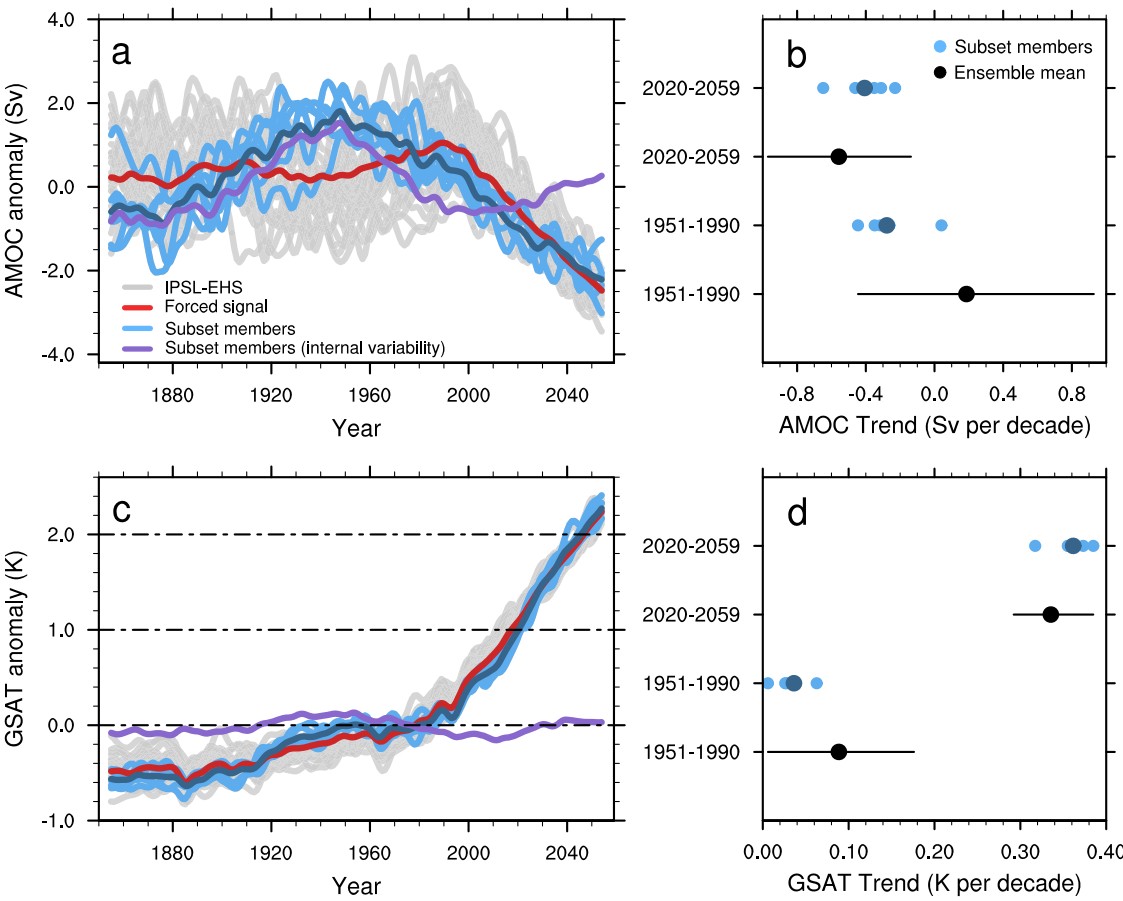

**Fig. 5 Implication for near-future AMOC and GSAT change. a** Time evolution of the low-pass filtered Atlantic Meridional Overturning Circulation (AMOC) anomaly (Sv) relative to the 1900–2018 period for the IPSL ensemble of extended historical simulations (IPSL-EHS) (gray), the ensemble mean (red), the subset of members identified in Fig. 2 (member #14, 5, 25, 29, 4 and 30 in light blue), the subset mean (dark blue) and the mean internal variability (calculated by removing the ensemble mean for each member) of the subset of members (purple). **c** Same as **a** but for the Global near-Surface Air Temperature (GSAT) anomaly (K). **b** AMOC trends (Sv per decade) in IPSL-EHS (black), with the ensemble mean (black dot) and the full minimum-maximum range (black line) and the subset of members (light blue), with the subset mean (dark blue). **d** Same as **c** but for GSAT.

## Methods

**Observational data**. We used the infilled Cowtan and Way[41,42], named HadCRUT4-CW in this study, observational surface temperature, which avoids the problem of missing values. This dataset is composed of blending sea surface temperature over the ocean and near-surface air temperature over the land. This metric of global mean temperature, however, warms significantly less than the GSAT. To take into account this discrepancy, we applied a factor of 1.06 to the HadCRUT4-CW surface temperature, as used in a recent study[10]. This factor is similar to those of 1.05[43], 1.06[44], and 1.08[5] used in several other studies. One of the interests of this dataset is to provide an ensemble of realizations, allowing to consider the associated uncertainties. Here, the median of the dataset is considered the best estimate, and the maximum and the minimum is used for the uncertainty. Note that the HadCRUT4-CW dataset is very consistent with the Berkeley[45] observational surface air temperature, with a Pearson correlation coefficient between the annual GSAT values of 0.99. Very small differences are found with the infilled HadCRUT5 dataset when accounting for the missing values of this dataset. Therefore, we keep the HadCRUT4-CW in order to have a global coverage without missing values.

**Simulations used**. The atmospheric resolution of IPSL-CM6A-LR is 1.26° × 1.25° with 79 levels (model top at 1 Pa). The model uses a nominal resolution of 1° in the ocean and 75 levels. The IPSL-CM6A-LR ensemble[12] used in this study follows the CMIP6 protocol[46] for historical simulations for the period 1850–2014. The historical simulations initial conditions are taken from different years in a long pre-industrial simulation after it has reached a quasi-stationary state. Specifically, the simulations are started from the atmospheric, oceanic, and land surface initial conditions of the 1st January from different years. The simulations were extended until 2060 using all forcings from the SSP245 scenario[47], except for the ozone field which has been kept constant to its 2014 climatology (as this particular forcing was not available at the time of performing the extensions). This implies that these simulations do not "see" the ozone hole recovery and changes in tropospheric

ozone, in contrast to the official CMIP6 projections. However, this is seen as a minor shortcoming for the purpose of this study. All the simulations used are summarized in Supplementary Table 1.

**ECS and TCR estimates**. The Sensitivity and the Transient climate response calculated over the historical period, named as S_hist and TCR_hist in this study, are defined as:

$$S\_hist = \frac{F_{2*CO2} * \Delta T}{\Delta F - \Delta Q} \tag{1}$$

$$TCR\_hist = \frac{F_{2*CO2} * \Delta T}{\Delta F} \tag{2}$$

with $F_{2*CO_2} = 3.77$ Wm$^{-2}$ being the radiative forcing due to a doubling of the atmospheric $CO_2$ concentration. This latter quantity is calculated by averaging the Effective Radiative Forcing (ERF) values from three different experiments performed with IPSL-CM6A-LR, a fixed SST experiment[48] and two experiments based on this method[49], using linear regressions of the top-of-atmosphere net radiative flux against surface temperature carried out over the first 20 years of the abrupt experiment, with a subtraction of the corresponding piControl quantities for both, but based on a year-to-year basis for the first case or using 20-year climatologies for the second case. We estimate the uncertainty related to using the lowest (3.50 W m$^{-2}$) and largest (3.94 W m$^{-2}$) values of the three experiments. The net ERF is calculated between the pre-industrial (1850–1879) and present-day (1999–2018) periods by averaging the ERFs of the three members of piClim-histall RFMIP experiments and is estimated at $\Delta F = 2.09$ Wm$^{-2}$. $\Delta T$ is the change in GSAT between the same periods, and $\Delta Q$ the rate of the total increase in Earth system heat content as diagnosed in the model, calculated over the 1999–2018 period. Here, we approximate the Earth System Heat Content by calculating $\Delta Q$ from the Ocean Heat Content increase over the historical period divided by 0.89, as the oceans store about 89% of the excess heat[50]. This allows us to take into account an

estimation of the continental and atmospheric heat uptake, as well as the heat uptake owing to ice melt. $\Delta F$ does not change much across the members so we use a single value. However both $\Delta T$ and $\Delta Q$ vary more substantially and are estimated separately for each member.

Although the piControl simulation inherits from a long millennial spin-up during the development process, the model is not completely equilibrated given the long timescale associated with the deep ocean. As a result, a very small drift is present in the piControl simulations, as it is often the case with climate models[51]. We diagnose the drifts in the piControl for OHC ($-1.46 \times 10^{21}$ J yr$^{-1}$) and GSAT ($-0.00011$ K yr$^{-1}$) and remove it in the historical simulations prior to the $\Delta Q$ and $\Delta T$ calculation.

**Calculation and analysis of the AMOC index**. The AMOC is evaluated using the maximum of the annual Atlantic meridional stream function at 20 °N–50 °N. We consider CMIP6 models with Atlantic meridional stream function data available in the piControl simulation for at least 500 years. Thirteen models, including IPSL-CM6A-LR, were available at the time of our analysis.

**Scaling of the internal versus externally-forced AMOC variation**. From the IPSL ensemble, we define the forced (or externally-driven) response of the AMOC as the ensemble mean of AMOC variations. This method is appropriate because the initial conditions of the 32 ensemble members started every 20 years from the piControl simulation to sample various phases of the multi-centennial variability. Then, for each member, the internal (or internally-driven) variability is obtained as the difference between that member and the forced response. To estimate the forced and unforced components in the AMOC strength variations for each member of the IPSL ensemble, we used a linear model as follow:

$$y_{AMOC\ unforced} = \lambda_{unforced}\, x_{AMOC} + \varepsilon_{unforced} \qquad (3)$$

$$y_{AMOC\ forced} = \lambda_{forced}\, x_{AMOC} + \varepsilon_{forced} \qquad (4)$$

where $x_{AMOC}$ is the same in both equations and designate the AMOC time series over the 1940–2016 period, $y_{AMOC\ unforced}$ the AMOC internal variability and $y_{AMOC\ forced}$ the AMOC forced signal while the $\lambda$ are the regression coefficients and $\varepsilon$ the error terms. When both $\lambda$ values are significantly positive, $\lambda$ estimates the respective roles of internal variability and forced changes in driving the simulated AMOC variations (e.g., Supplementary Fig. 1).

**Calculation of the AMV index**. The AMV index is calculated as the mean SST in the North Atlantic Ocean between 0 °N and 60 °N. How to remove the effect of external forcings in order to study the AMV due to internal processes from the observations and the historical simulations is a thorny issue. A good way to do so in ensemble simulations is to remove the ensemble mean. However, this cannot be done in observations for which we only have one realisation. Therefore, when comparing the simulated AMV to the observations, as in Fig. 3b, the forced signal is estimated from the average SST between 60 °S and 60 °N[52]. In both observation and model[12], the AMV spatial pattern is characterized by SST anomalies of the same sign over the North Atlantic, with a maximum on the subpolar gyre and a second maximum off the Iberian Peninsula and in the tropical Atlantic. A Lanczos low-pass filter with a cutoff period of 11 years is then used to retain only the low-frequency variations.

**Significance testing**. The significativity of the relationship between the AMOC trends and the TCR_hist and TCR_hist and the regression calculated in Fig. 2a are estimated with a two-tailed Student t-test considering a $p$-value < 0.1 and 30 degrees of freedom. For the significance of the regression between GSAT and AMOC trends in the piControl simulations (Fig. 4a), we used a nonparametric method[53] based on a random phase of resampling, using 1000 surrogates and $p$-value < 0.1, in order to take into account the serial correlation, as the trends overlap. The same method is used to evaluate the significativity of the correlations between AMV and AMOC time series in IPSL-EHS and between the four observed AMOC fingerprints and the member #14 in the third section.

## Data availability
All of the observational datasets used are publicly available online. The CMIP6 model outputs from the piControl and historical simulations are available through the Earth System Grid Federation (ESGF) portal (https://esgf-node.llnl.gov/projects/cmip6/). The data based on the extended (beyond 2014) historical simulations from the IPSL-CM6A-LR model used in the main figures of this study are available in this repertory: https://doi.org/10.5281/zenodo.5159426.

## Code availability
The codes used for the analyses are available from the authors upon request.

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

## Acknowledgements

The IPSL-CM6 experiments were performed using the HPC resources of TGCC under the allocations 2017-R0040110492, 2018-R0040110492, and 2019-A0060107732 (project gencmip6) provided by GENCI (Grand Equipement National de Calcul Intensif). The IPSL-CM6 team is acknowledged for data curation of the CMIP6 model output. RB and OB were supported by the European Union's Horizon 2020 research and innovation programme under grant agreement number 820829 for the "Constraining uncertainty of multi-decadal climate projections (CONSTRAIN)" project (07/2019-06/2024). JM, GG, and DS were supported by Blue-Action (European Union's Horizon 2020 research and innovation program under grant agreement no 727852) and EUCP (European Union's Horizon 2020 research and innovation programme under grant agreement no 776613) projects. This work also benefited from the French state aid managed by the ANR under the "Investissements d'avenir" programme with the reference ANR-11-IDEX-0004-17-EURE-0006.

## Author contributions

R.B., D.S., O.B., and J.M. designed the study. R.B. performed all of the IPSL-EHS and the multi-model analysis against observations. G.G. computed the A.M.O.C. index based on the O.H.C. AS provided the radiative forcing estimates. RB wrote the paper, and O.B., D.S., J.M., J.D., G.G., F.H., and J.S. contributed to the text.

## Competing interests

The authors declare no competing interest.
