## [Peer Review File · Nature Communications]

REVIEWER COMMENTS

Reviewer #1 (Remarks to the Author):

Review of "Increased risk of near term global warming level due to a recent AMOC weakening"

The study shows impact of internal variability on the rate of global surface air temperature increase (GSAT) in an of ensemble of the historical simulations with the IPSL. They showed that the GSAT trends are correlated with AMOC trends among the ensemble members, and the ensemble members that reproduced the observed AMOC trend also reproduce the observed GSAT. They further demonstrate this relation using different AMOC fingerprints, and further argue that the relationship holds for other CMIP6 models that significant centennial variability of the AMOC. I find the study to well written and the arguments are logical and mostly supported by the figures. I recommend the paper for publishing after major revision.

Major comments:

1. The introduction of the study is motivated entirely by the relatively higher than observed climate sensitivity among the CMIP6 models and the implications of the study is discussed in terms of these quantities. However, the results are presented in terms of the GSAT changes. While the study clearly shows that GSAT trends of the subset ensemble members selected are related to the AMOC trends, the relationship may not hold for S_hist and TCR and the AMOC trend among the ensemble. At least for the 6 selected ensemble members, only 3 of them have S_hist values that are below the ensemble average, the other three have sensitivities greater than the ensemble average according by the figure 1b. Care should be taken on how the result here are interpreted in terms of climate sensitivity (S_hist and TCR). Perhaps a plot similar to figure 2a for S_hist and T_CR will make this interpretation more clear
2. The assumption that AMOC trend in observations is largely internally driven is not very well supported in the study. Though this is shown for the selected ensemble members, it is not necessarily true for observation. More references can be given to support this. Given the large correlation between the observed AMOC and the AMV, Steinman et al 2015 and Qin et al 2020 are useful references have shown that this assumption is valid, especially for the late 20th century period.
3. I would include figure S9 in the main text and move figure 1 to the supplements. It would be useful to show the internal variability of the remaining ensemble members in figure S9 (similar to the purple line) to show how the internal variability differs for the rest of the ensemble members.

Minor comments

138-141: The AMOC impact on surface temperatures is also clearly demonstrated in the idealized study of Garuba et al 2018.

209-211: The sentence is not very clear please reword.

235-238: It is not mentioned if these correlations are statistically significant.

265-269: It will be helpful to mention some of the models being referred to here.

References

- Steinman, B. A., Mann, M. E., & Miller, S. K. (2015). Atlantic and Pacific multidecadal oscillations and Northern Hemisphere temperatures. *Science*, 347, 988–991.
<https://doi.org/10.1126/science.1257856>
- Garuba, O. A., Lu, J., Liu, F., & Singh, H. A. (2018). The active role of the ocean in the temporal evolution of climate sensitivity. *Geophysical Research Letters*, 45, 306– 315.

<https://doi.org/10.1002/2017GL075633>

Qin, M., Dai, A., & Hua, W. (2020). Quantifying contributions of internal variability and external forcing to Atlantic multidecadal variability since 1870. *Geophysical Research Letters*, 47, e2020GL089504. <https://doi.org/10.1029/2020GL089504>

Reviewer #2 (Remarks to the Author):

Review of "Increased risk of near term global warming due to a recent AMOC weakening" by Bonnet et al.

Overall this is an interesting and topical paper, making the argument that a combination of a high climate sensitivity and negative AMOC trend may be consistent with observed recent climate evolution, and could result in enhanced warming in the future. This is an interesting a provocative subject, and overall this is worthy of publication in my view, with some revision.

There are some other relevant ideas and topics in the literature which might be referenced to better set these ideas in context. In particular, the hypothesis proposed is related to, but distinct from, the hypothesis that internal variability associated with pattern effect in Pacific SST has led to a weaker warming in recent decades, and therefore that a higher climate sensitivity and strong future warming are more likely (see e.g. Zhou et al., 2021 - <https://www.nature.com/articles/s41558-020-00955-x>). Both hypotheses suggest that internal variability may have masked part of the human-induced global warming in recent decades, and hence suggest stronger future warming.

The study should also better reference other literature reporting high multi-decadal variability in CMIP6 models, for example, this study by Parsons et al., <https://doi.org/10.1029/2019GL086588>, and the Ribes et al. (2021) which is already referenced, but not on this point.

On line 120, the authors correctly note that their conclusions depend on the multi-decadal variability in IPSL-CM6A-LR being realistic. And on lines 276-277 they suggest that the stronger multidecadal variability in this and other CMIP6 models may be more consistent with paleodata. But Parsons et al., Figure 2a, middle panel, shows that the CMIP6 models with the highest interdecadal variability in GMST, of which this model is one, have variability exceeding that in proxy reconstructions over the 1450-1840 period. The authors should comment on this and include more evaluation and literature assessment of decadal variability in this model.

Specific comments:

Ln 13: Replace "Part of the new generation of CMIP6 models" with "Some of the new generation CMIP6 models"

Ln 32: Susceptibility -> sensitivity

Ln 32-33: Some models from CMIP6 have higher ECS, but not all models.

Ln 36: It is not true that CMIP6 models consistently project a greater warming than projected by CMIP5. Some CMIP6 models project greater warming, but not all.

Ln 50-54: It is not true that Tokarska et al. neglect the contribution of internal variability – they write this: 'Internal variability can also affect the warming rate. Pacific variability has temporarily slowed short-term warming during the "global warming hiatus" from the late 1990s up to around year 2012 (23, 24)...'. Moreover they devote two whole paragraphs of their methods section to how they account for internal variability in their results, including approaches to accounting for Atlantic and Pacific decadal variability. Also Liang et al., 2021

(<https://doi.org/10.1029/2019GL086757>) is another study which explicitly accounts for internal variability in deriving observationally-constrained warming projections.

Ln 87: The argument made here does not seem to be a convincing refutation of the hypothesis that the too strong warming is due to too high radiative forcing. The authors would have to show this more quantitatively to reach such strong conclusions. Or they could keep the evidence the same, but draw a weaker conclusion.

Ln 127-128: It isn't clear what an 'ascending phase' of the AMOC means. From the context, the authors seem to mean a 'strong phase'. But 'ascending phase' made me think of rising water in the N Atlantic rather than sinking water. Also can the authors give a brief explanation for why a strong AMOC should lead to accumulation of fresh water in the Arctic?

Ln 130-131: It isn't clear why a larger sea ice loss should result from an accumulation of fresh water near the surface of the Arctic Ocean – I would expect the reverse. Can the authors explain this better?

Ln 135-138: My understanding is that there is substantial uncertainty in these reconstructions – perhaps the authors could add some additional brief discussion of the uncertainties here.

Ln 267-268: Say here how these 13 models were selected.

Ln 275-277: See Parsons et al. (2021) – there is evidence that some CMIP6 models may overestimate multi-decadal variability in GMST compared to proxy reconstructions.

Ln 297: What is the basis for concluding that CanESM5 has too weak internal variability? Or is it omitted just because none of its ensemble members reproduce observed warming?

Ln 315: Where is it shown that these models have low frequency variability which is in disagreement with observations?

Ln 340-341: Liang et al., (2021) do use individual ensemble members as suggested here.

Ln 453-454: Rather than using HadCRUT4-CW, which is a version of HadCRUT4 infilled after the development, the authors could use HadCRUT5, which was infilled by the team which developed the dataset, and has a number of other improvements.

Ln 505: 'inherits' -> 'starts'

Ln 522-528: This description of the regression method isn't clear to me. Some symbols seem to be missing from the PDF, for example before 'estimates' on line 527. Are both instances of xAMOC in the two equations the same thing? Overall this doesn't make sense to me. Clarify.

Reviewer #3 (Remarks to the Author):

I did not realize that this is the manuscript which I have reviewed before for Nature Climate Change. Although this manuscript is only slightly modified from the last version submitted to Nature Climate Change, to me, the findings in this manuscript are important and the manuscript is worth to be published by Nature Communication.

In this manuscript, the authors studied the potential influence of the internal variability portion of the AMOC on the simulated global mean surface temperature (GSAT) change during 1946-2018 and the implication to the relationship between observed AMOC and GSAT. They found that the weakening trend of the simulated AMOC since 1940s may have contributed to the closer to observed GSAT changes since a weaker AMOC induces a cooling effect in the Northern Hemisphere and a lessened GSAT warming. They also suggest that the ECS constrained by observations in the 20th century in previous studies may have underestimated by the possible weakening of AMOC in observations in this period. To me, the authors have done very careful analysis and comparison of the observed and model simulated data, and the conclusions reached here are also well demonstrated.

Minor comments:

1. It is worth to explore the magnitude of the GSAT change in relation to AMOC changes in other CMIP6 models to see whether magnitude of GSAT change per 1 Sv AMOC change in IPSL-CM6A-LR model is above, below the other CMIP6 model. By doing so, one may have a sense to see whether IPSL model has significantly over or underestimated the AMOC's effect on GSAT.
2. How the multi-centennial AMOC variability in historical runs differ from that in the control run? Whether the members with an AMOC declining trend are affected by the AMOC initial state when these members are branched from the control run.

Response to reviews of “Increased risk of near term global warming level due to a recent AMOC weakening” by Bonnet et al.

We want to thank the three Reviewers for their careful reading and for their comments and suggestions that helped us to improve the clarity and the strength of our study. The main comments from the Reviewers revolve around: i) the realism of the low-frequency internal climate variability in the IPSL-CM6A-LR model, ii) some clarifications about the interpretation of S_hist and TCR_hist, iii) the use of HadCRUT4-CW and iv) referencing and discussing previous studies. The main changes made to the manuscript are the following: i) Fig. S9 has been added to the main text, ii) Fig. R4 has been added in the supplementary, iii) a number of the references suggested by the Reviewers have been cited and discussed, iv) discussion about the low-frequency internal climate variability of IPSL-CM6A-LR has been added and v) clarifications about the S_hist and TCR_hist interpretation has been made. For clarity, the Reviewers' comments are in **bold**.

Reviewer #1 (Remarks to the Author):

The study shows impact of internal variability on the rate of global surface air temperature increase (GSAT) in an of ensemble of the historical simulations with the IPSL. They showed that the GSAT trends are correlated with AMOC trends among the ensemble members, and the ensemble members that reproduced the observed AMOC trend also reproduce the observed GSAT. They further demonstrate this relation using different AMOC fingerprints, and further argue that the relationship holds for other CMIP6 models that significant centennial variability of the AMOC. I find the study to well written and the arguments are logical and mostly supported by the figures. I recommend the paper for publishing after major revision.

We thank the Reviewer for his/her positive evaluation.

Major comments:

1. The introduction of the study is motivated entirely by the relatively higher than observed climate sensitivity among the CMIP6 models and the implications of the study is discussed in terms of these quantities. However, the results are presented in terms of the GSAT changes. While the study clearly shows that GSAT trends of the subset ensemble members selected are related to the AMOC trends, the relationship may not hold for S_hist and TCR and the AMOC trend among the ensemble. At least for the 6 selected ensemble members, only 3 of them have S_hist values that are below the ensemble average, the other three have sensitivities greater than the ensemble average according by the figure 1b. Care should be taken on how the result here are interpreted in terms of climate sensitivity (S_hist and TCR). Perhaps a plot similar to figure 2a for S_hist and T_CR will make this interpretation more clear.

Indeed, we strongly insist on climate sensitivity in the introduction because (i) this is a crucial quantity to estimate to cope with targets like the Paris agreement, (ii) there is a strong debate on why some CMIP6 models do have stronger climate sensitivity than CMIP5 models and how realistic this might be (iii) this quantity is mainly constrained by observations of the

recent decades, where sufficient data are available, so that (iv) its estimation over this time period can be strongly affected by low frequency climate variability in some models as shown in this study.

Nevertheless, the Reviewer is correct that for the 6 selected members (#4, 5, 14, 25, 29, and 30), 3 have apparent ECS (S_{hist}) smaller than the ensemble mean and the other 3 have S_{hist} larger than the ensemble mean. However this is not true for TCR_{hist} , with 4 members having a TCR_{hist} smaller than the ensemble mean, 1 member slightly above, and 1 member larger than the ensemble mean. The main message from the Fig. 1b and c is that the internal variability has a strong influence on the estimation of S_{hist} and TCR_{hist} and that these values are on average lower than the estimate from the abrupt4xCO2 experiment. Consequently it seems difficult to estimate those quantities over the historical period from observations. This seems to be a key preliminary result from our manuscript. At this stage of the manuscript, we are not arguing that the AMOC explains the spread over this time period. It is only in a second stage that we show that the members most in line with GSAT are those showing strong AMOC decrease since 1950. Thus, from this, we can mainly conclude that S_{hist} defined as the difference between the pre-industrial (1850-1879) and present-day (1999-2018) periods might be very misleading, due to potential very low frequency variability in the climate system. We also conclude that other definitions accounting for internal variability might be more relevant to correctly estimate this quantity. The same is true for TCR_{hist} .

As anticipated by the Reviewer, there is no strong relationship between S_{hist} or TCR_{hist} and the AMOC trend among the ensemble over the 1940-2016 period (Figure R1 a-b). This is due to the fact that these quantities rely on the temperature difference between the pre-industrial (1850-1879) and present-day (1999-2018) periods while the relationship found Fig. 2a is for trends calculated over the 1940-2016 period, when the AMOC is suspected to decrease in observations. It can be noted that the relationship between S_{hist} or TCR_{hist} and the AMOC trend is stronger and significant when considering the whole historical period (1850-2018) (Figure R1 c-d).

In conclusion, we keep the question of the climate sensitivity in the framing of the paper, but we added a sentence in the revised manuscript to point to the fact that the S_{hist} and TCR_{hist} do not have a clear relationship with the AMOC trends over the 1940-2016 period. We also deleted the point in the abstract on the observational estimate of the climate sensitivity which could be confusing. While this is worth noting, this does not invalidate the conclusions of our study.

Figure R1: **a**, Scatter plot of the S_hist (in K) versus the AMOC trends (Sv per decade) calculated over the 1940-2016 period and for each individual member of the IPSL ensemble. The black solid line represents the least square regression between these two variables in the ensemble. **b**, Same as **a** but for TCR_hist. **c** and **d**, Same as **a** and **b**, but with the AMOC trends calculated over the whole historical period (1850-2018). The determination coefficients and the p-value based on a two-tailed Student t-test are indicated in the bottom right.

2. The assumption that AMOC trend in observations is largely internally driven is not very well supported in the study. Though this is shown for the selected ensemble members, it is not necessarily true for observation. More references can be given to support this. Given the large correlation between the observed AMOC and the AMV, Steinman et al 2015 and Qin et al 2020 are useful references have shown that this assumption is valid, especially for the late 20th century period.

We thank the Reviewer for pointing out these useful studies. In our study, we proposed a scaling of the internal versus externally-forced AMOC variation. In this respect, Fig. S1 is providing compelling evidence of the fact that in this model (which is the main limitation of the present attribution), the AMOC trend since 1940 is mainly driven by internal variability, and this is not only true in the subset showing a weakening, but also in the other members

(even when they are not showing any weakening). The important role of the internal variability in the AMOC variations over the historical period is also clearly visible in Figure 2b. This might be due to the fact that external forcing is playing a strong role in AMOC weakening only since the last one or two decades in historical simulation, as highlighted in the whole CMIP6 database by Menary et al. (2020), while anthropogenic aerosol forcing on the opposite might have forced a slight increasing trend.

We agree with the Reviewer that these analyses are only based on one ensemble model, which limits the strength of our hypothesis. As there are no available long-term observations of the AMOC, it is necessary to rely on indirect observations of the AMOC, like the AMV for example, but we also used three other AMOC fingerprints in the study in order to reinforce this hypothesis.

As suggested by the Reviewer, a point about the debate concerning the origin and process governing the AMV, which might be due to the imprints of external forcing and/or internal climate variability is now added in the revised manuscript, including the Steinman et al. (2015) and the Qin et al. (2020) references.

3. I would include figure S9 in the main text and move figure 1 to the supplements. It would be useful to show the internal variability of the remaining ensemble members in figure S9 (similar to the purple line) to show how the internal variability differs for the rest of the ensemble members.

Indeed, we agree that putting Figure S9 in the main text will better highlight the results. However, we think that the Figure 1 is also important, as it highlights the fact that despite the larger than observed ensemble mean of the IPSL model, some members have a good consistency with the observations, but also the fact the S_hist and TCR_hist are largely influenced by the internal climate variability. In conclusion, we kept Figure 1 in the main manuscript and Figure S9 is now Figure 5 in the revised manuscript.

Minor comments

138-141: The AMOC impact on surface temperatures is also clearly demonstrated in the idealized study of Garuba et al 2018.

Thank you for this reference that we were not aware of. It is now added to the revised manuscript.

209-211: The sentence is not very clear please reword.

We have modified this sentence to make it clearer in the revised manuscript. The new sentence reads: "The upper OHC is indeed significantly correlated with the AMOC in IPSL-CM6A-LR over these two regions, with a negative correlation offshore Newfoundland and a positive correlation over the North Atlantic Subpolar Gyre (Fig. S5a). This dipole in correlation is consistent with observations and other climate model simulations."

235-238: It is not mentioned if these correlations are statistically significant.

It was indicated in lines 237-238 that these correlations are statistically significant for three of the four AMOC fingerprints used: the Caesar index, the Δ OHC index and Δ ITA index.

265-269: It will be helpful to mention some of the models being referred to here.

We have now mentioned the models with a significant relationship between the AMOC and GSAT 60 years trends in their piControl simulation.

References

Steinman, B. A., Mann, M. E., & Miller, S. K. (2015). Atlantic and Pacific multidecadal oscillations and Northern Hemisphere temperatures. *Science*, 347, 988–991.

<https://doi.org/10.1126/science.1257856>

Garuba, O. A., Lu, J., Liu, F., & Singh, H. A. (2018). The active role of the ocean in the temporal evolution of climate sensitivity. *Geophysical Research Letters*, 45, 306– 315.

<https://doi.org/10.1002/2017GL075633>

Qin, M., Dai, A., & Hua, W. (2020). Quantifying contributions of internal variability and external forcing to Atlantic multidecadal variability since 1870. *Geophysical Research Letters*, 47, e2020GL089504.

<https://doi.org/10.1029/2020GL089504>

Reviewer #2 (Remarks to the Author):

Overall this is an interesting and topical paper, making the argument that a combination of a high climate sensitivity and negative AMOC trend may be consistent with observed recent climate evolution, and could result in enhanced warming in the future. This is an interesting and provocative subject, and overall this is worthy of publication in my view, with some revision.

We thank the Reviewer for his/her positive evaluation.

There are some other relevant ideas and topics in the literature which might be referenced to better set these ideas in context. In particular, the hypothesis proposed is related to, but distinct from, the hypothesis that internal variability associated with pattern effect in Pacific SST has led to a weaker warming in recent decades, and therefore that a higher climate sensitivity and strong future warming are more likely (see e.g. Zhou et al., 2021 - <https://www.nature.com/articles/s41558-020-00955-x>). Both hypotheses suggest that internal variability may have masked part of the human-induced global warming in recent decades, and hence suggest stronger future warming.

Indeed Zhou et al. (2021) show that the pattern effect could have masked part of the human-induced global warming in recent decades. This pattern effect could be linked to the non-uniform warming in the tropics (Fueglistaler, 2019) or internal variability with the contribution of the Pacific Ocean variability modulating the heat uptake over the recent decades (as

reviewed in Xie et al. 2020). Zhou et al. (2021) do not specifically attribute their estimated pattern effect to a particular reason. Hence we agree with the Reviewer that our hypothesis relates to but is distinct from that of Zhou et al. (2021). Here, we rather focus on the internal variability of the Atlantic Ocean circulation, which could also lead to a pattern effect. We have added a reference to Zhou et al. (2021) in our revised manuscript:

“These results thus are in line with a recent study (Zhou et al., 2021) suggesting that the pattern effect could have masked part of the human-induced global warming in recent decades, although the time period we investigate here is longer.”

The study should also better reference other literature reporting high multi-decadal variability in CMIP6 models, for example, this study by Parsons et al., <https://doi.org/10.1029/2019GL086588>, and the Ribes et al. (2021) which is already referenced, but not on this point.

Indeed, we referred to the study by Ribes et al. (2021) but on a different point. We have added a short discussion about the relatively high multi-decadal variability in some CMIP6 models in the introduction of the revised manuscript. Please find below the new paragraph:

“This new generation of models with large effective ECS raises important questions: i) are these models realistic in comparison to present-day and climate change observations? ii) how to interpret these highly sensitive models in relation to historical climate change? If these models are not falsifiable, it would imply higher risks and costs induced by future climate change impacts and the need for greater and faster mitigation efforts to achieve climate targets than previously thought. To address these issues, recent studies (Tokarska et al., 2020; Nijse et al., 2020; Liang et al., 2020) constrain CMIP6 climate model projections with observed warming trends over the last decades. Although it is interesting to focus on the last few decades due to large observational evidence availability, it is important to keep in mind that low-frequency variability, as found in some of CMIP6 models, might also contribute to observed multi-decadal trend and can strongly hamper our possibility of estimating the exact response to external forcings. In order to take into account all the information available over the historical period, a recent study (Ribes et al., 2021) developed a new statistical method, leading to a reduction of the uncertainty on the projected future warming by about 50% and a slightly higher future warming in CMIP6 relative to CMIP5. However, this method has some difficulties to capture the main features of the internal variability simulated by the CMIP6 models (Parsons et al., 2020; Ribes et al., 2021) with higher low-frequency internal climate variability than previous model-based estimations with potentially larger impacts. Indeed, the low-frequency internal variability at decadal to multi-centennial time scales can temporarily enhance or reduce the long-term effects of climate change. Such variability might also affect the way climate models are compared to observations.”

On line 120, the authors correctly note that their conclusions depend on the multi-decadal variability in IPSL-CM6A-LR being realistic. And on lines 276-277 they suggest that the stronger multidecadal variability in this and other CMIP6 models may be more consistent with paleodata. But Parsons et al., Figure 2a, middle panel, shows that the CMIP6 models with the highest interdecadal variability in GMST, of which this model is one, have variability exceeding that in proxy reconstructions over the 1450-

1840 period. The authors should comment on this and include more evaluation and literature assessment of decadal variability in this model.

We thank the Reviewer for raising this point which is indeed key for our study. Fig. 2a from Parsons et al. (2021) shows that interdecadal variability is larger in CMIP5 models (and probably by extension in CMIP6 models) than in paleoclimate proxies over the 1450-1849 period after removing trends at the grid box level. We acknowledge that the IPSL-CM6-LR is on the high side in terms of interdecadal GSAT variability, with a standard deviation of 0.12°C in the *piControl* simulation in comparison to the paleoclimate proxies used in the study for which a standard deviation of around 0.07°C for period larger than 25-yr is reported by the authors for the 1450-1850 low-forcing period. Nevertheless, the Parsons et al. (2021) study is focusing on interdecadal variability (notably since it is only comparing model with 400-year long reconstruction, which is still a bit short to assess multi-centennial variability), while in our study, this is rather the multi-centennial variability that play a crucial role. In that respect, Parsons et al. (2021) is not allowing to make any proper assessment of our simulated internal variability.

Furthermore, for these multi-centennial time scales uncertainties in proxy-based reconstructions remain too large to make a proper evaluation of the amplitude of the low-frequency variability from climate simulations (cf. Fig. 2a from PAGES2K 2019, or Moberg et al. 2005). Additionally, there are also large uncertainties in the reconstructed forcing used in the millennium simulation, as well as in the climate response (Swingedouw et al. 2017). Such analyses are nothing but easy and necessitate a dedicated study.

The historical period 1850-present can also be used to quantify the internal climate variability using different methods to estimate the forced part of the climate. In the IPSL-CM6A-LR model, a large part of the low-frequency internal climate variability of the GSAT is related to the AMOC and the AMV. Several studies have shown that internal variability related to the AMV or the PDV is in general too weak and not persistent enough in comparison to the observations (Cheung et al., 2017; Qasmi et al., 2017; Peings et al., 2016). Therefore, from this point of view, IPSL-CM6A-LR could be more consistent with the observations.

Nevertheless, we agree with the Reviewer that this point needs more discussion in the manuscript. Therefore, we added a new paragraph in the last section of the revised manuscript about it. The new paragraph reads:

“The realism of the multi-centennial low-frequency internal variability found in some of the CMIP6 models (Parsons et al., 2020; Ribes et al., 2021) is a crucial element of our results. Although the instrumental period is short, previous studies suggested that the Atlantic Multidecadal Variability is underestimated in CMIP5 models, with a persistence lower than the one deduced from observations (Cheung et al., 2017; Qasmi et al., 2017). To gain insight on this realism, paleodata are useful but usually suffer from large uncertainties concerning quantified estimations of the variability, especially at the large scale. While it is usually believed that model simulations might have too low multi-centennial variability as compared to proxy records (Laepple and Huybers, 2014), a recent study (Parsons et al., 2020) suggests instead that GSAT interdecadal variability of some CMIP5 models might be overestimated as compared to PAGES2K reconstructions over the period 1450-1840. On the longer time scale of the last two thousand years, the different methods to reconstruct global

mean surface temperature do show considerable uncertainty in terms of amplitude of multi-centennial variability (PAGES2K, 2019), so that it is difficult at this stage to properly assess the realism of model simulations against those reconstructions. Such an evaluation of model simulations might deserve a dedicated analysis using PAGES2K database and last millennium simulations from CMIP6 archives, using advanced techniques like pseudo-proxy approaches (Ortega et al., 2015) in order to compare model simulations and reconstructions in a coherent framework.”

Specific comments:

Ln 13: Replace “Part of the new generation of CMIP6 models” with “Some of the new generation CMIP6 models”

Done.

Ln 32: Susceptibility -> sensitivity

Done.

Ln 32-33: Some models from CMIP6 have higher ECS, but not all models.

We clarified this point in the revised manuscript.

Ln 36: It is not true that CMIP6 models consistently project a greater warming than projected by CMIP5. Some CMIP6 models project greater warming, but not all.

We agree with the Reviewer. We referred here to the part of the models with larger ECS than the IPCC AR5 likely range, as referred on lines 35-36. We modified the sentence to avoid confusion.

Ln 50-54: It is not true that Tokarska et al. neglect the contribution of internal variability – they write this: ‘Internal variability can also affect the warming rate. Pacific variability has temporarily slowed short-term warming during the “global warming hiatus” from the late 1990s up to around year 2012 (23, 24)...’. Moreover they devote two whole paragraphs of their methods section to how they account for internal variability in their results, including approaches to accounting for Atlantic and Pacific decadal variability. Also Liang et al., 2021 (<https://doi.org/10.1029/2019GL086757>) is another study which explicitly accounts for internal variability in deriving observationally-constrained warming projections.

We agree that this sentence was misleading. As suggested by the Reviewer, we have modified this part of the text in order to avoid any misunderstanding and better introduce our work in the context of previous studies. We also thank the Reviewer for pointing us to the study of Liang et al. (2021), which is now added to the revised manuscript.

It should be noted that we did not say that Tokarska et al. neglect the contribution of internal variability but rather that it was not fully accounted for. Our understanding of the Tokarska et al. study is that internal variability is only taken into account statistically and not on the longer timescales (e.g. centennial) that this study is concerned about.

Ln 87: The argument made here does not seem to be a convincing refutation of the hypothesis that the too strong warming is due to too high radiative forcing. The authors would have to show this more quantitatively to reach such strong conclusions. Or they could keep the evidence the same, but draw a weaker conclusion.

We agree that our statement made here was too strong and we have modified the sentence accordingly. Please find below the new sentence.

“While the radiative forcing by aerosols in IPSL-CM6A-LR is relatively weak at -0.6 Wm^{-2} , which might be in favor of hypothesis (ii) we argue that this is compensated by an underestimation of the (positive) radiative forcing by non-CO₂ greenhouse gases (Boucher et al., 2020). Thus, although we cannot discard hypothesis ii) in the light of the large uncertainties on the aerosol forcing, we focus this study on the exploration of hypotheses i) and iii).”

Ln 127-128: It isn't clear what an 'ascending phase' of the AMOC means. From the context, the authors seem to mean a 'strong phase'. But 'ascending phase' made me think of rising water in the N Atlantic rather than sinking water. Also can the authors give a brief explanation for why a strong AMOC should lead to accumulation of fresh water in the Arctic?

By 'ascending phase', we mean a positive anomaly of the AMOC (i.e. a strengthening of the AMOC). We rewrote this paragraph in the revised manuscript in order to clarify this point, as well as to add some information about the mechanisms behind it. Please find below the new paragraph.

“IPSL-CM6A-LR is characterized by a multi-centennial variability of AMOC and GSAT in the *piControl* simulation (Boucher et al., 2020). A recent study (Jiang et al., 2021) suggests that a positive AMOC anomaly is associated with warm temperature anomalies in the Northern Hemisphere, a reduced Arctic sea-ice extent, and salinity-driven positive density anomalies in the Nordic Seas and in the North Atlantic subpolar gyre. During a positive AMOC, both ocean and sea ice freshwater export from the Arctic Ocean decreases, which contributes to build a large freshwater anomaly in the top 100m of the Arctic Ocean, while the Atlantic inflow of salty water into the Arctic increases the salinity below 100m. After several decades, the freshwater anomaly is covering most of the western Arctic which eventually increases the oceanic freshwater export from the Arctic into the Nordic Seas through the Fram Strait. This slows down the AMOC. The Northern Hemisphere climate then cools and Arctic sea ice grows, reversing the phase of the cycle. The 32 members of the ensemble were run with the same external forcings but initialised from different years in the pre-industrial control simulation in order to sample this multi-centennial variability. Due to the link between the AMOC variability and the initial state use in the historical simulations (Fig S4), a clear relationship exists between the AMOC trend in 1850-2010 and the *S_hist* and *TCR_hist* (not shown).”

Ln 130-131: It isn't clear why a larger sea ice loss should result from an accumulation of fresh water near the surface of the Arctic Ocean – I would expect the reverse. Can the authors explain this better?

The melting of sea ice induced by a larger AMOC modifies the freshwater exchanges at the ice / ocean interface, with less freshwater ice export from the Arctic through Fram strait. The ocean also changes in parallel, with oceanic freshwater transport anomalies lagging the AMOC by a few decades. As indicated in the response above, we modified this paragraph in order to better describe the mechanisms at play. Please find the new paragraph in the previous comment.

Ln 135-138: My understanding is that there is substantial uncertainty in these reconstructions – perhaps the authors could add some additional brief discussion of the uncertainties here.

We agree that substantial uncertainty exists concerning this reconstruction, and this is the reason why we considered additional fingerprints to avoid basing all the analysis only on this index. Nevertheless, this index is also supported by a few additional lines of observational evidence (cf. Caesar et al., 2021), although not all the other reconstructions agree on the weakening trend at the moment. We have specified this in the revised manuscript.

“Although some debate exists concerning the robustness of this index, the implied AMOC weakening trend is supported by other reconstructions and indices (Caesar et al., 2021). There are still substantial uncertainties though, and other AMOC fingerprints will be considered later on in this study. We start by using the simulated AMOC, without any measurement uncertainties.”

Ln 267-268: Say here how these 13 models were selected.

This was indicated in the method section of the original manuscript: “We consider CMIP6 models with Atlantic meridional stream function data available in the *piControl* simulation for at least 500 years. Eight models, including IPSL-CM6A-LR, were available at the time of our analysis.”

Ln 275-277: See Parsons et al. (2021) – there is evidence that some CMIP6 models may overestimate multi-decadal variability in GMST compared to proxy reconstructions.

Please see our response above.

Ln 297: What is the basis for concluding that CanESM5 has too weak internal variability? Or is it omitted just because none of its ensemble members reproduce observed warming?

CanESM5 is omitted because none of its ensemble members reproduce the observed warming. The text was changed to “First, one can see that some members in the CNRM and the MPI ensembles reproduce the observed warming (similar GSAT trend as observations),

unlike the CanESM5 ensemble. This latter model is therefore omitted. This may be due to its large climate sensitivity and/or its weak internal climate variability.”

Ln 315: Where is it shown that these models have low frequency variability which is in disagreement with observations?

We agree the conclusion was not well substantiated and that our sentence was not clear. Indeed, only the CanESM5 ensemble is not consistent with the observed trend. This paragraph has been rewritten to make it clearer. Please find below the new paragraph.

“In conclusion the analysis of three other large ensembles of *historical* simulations does not invalidate our results that suggest that the weakening in AMOC reconstructions since the second half of the 20th century might be due to internal variability. Indeed, a similar behaviour can be seen in one model (CNRM-CM6 with a high ECS), the CanESM5 model does not reproduce the IPSL-CM6A-LR features as it appears to have a warming not compatible with the observations, and the MPI-ESM1.1 model, in which the members with an AMOC trend consistent with a recent reconstruction (Caesar et al., 2018) are not those with the best representation of the four AMOC fingerprints analyzed in this study.”

Ln 340-341: Liang et al., (2021) do use individual ensemble members as suggested here.

Thank you for the reference, please also refer to our response above.

Ln 453-454: Rather than using HadCRUT4-CW, which is a version of HadCRUT4 infilled after the development, the authors could use HadCRUT5, which was infilled by the team which developed the dataset, and has a number of other improvements.

We choose the HadCRUT4-CW (Cowtan and Way, 2014) dataset because it reports infilled temperature without any missing values over the historical period, which represents the largest contribution to the discrepancy when comparing the Global Mean Surface Temperature (GMST) from observations to the GSAT metric from models (Gillett et al., 2021).

Figure R2 below shows a comparison between the HadCRUT4-CW and the HadCRUT5 dataset, the latter having a slightly larger warming. In the infilled version of HadCRUT5, there are still missing values during the historical period. Very small differences are found between the HadCRUT4-CW and the HadCRUT5 data when accounting for missing values. Consequently, we keep the HadCRUT4-CW dataset in order to have a complete coverage. Note that we used a scale factor of 1.06 in order to correct the differences between the blended data of the observations and the GSAT from the models as done in several other studies (Cowtan and Way, 2015; Richardson et al., 2016; Gillett et al., 2021). A sentence was added in the first section of the Methods part of the revised manuscript.

Figure R2: Comparison of annually- and globally-averaged temperature anomaly series (°C) from the HadCRUT5 dataset (black), the HadCRUT4-CW dataset (green) and the HadCRUT4-CW dataset masked with HadCRUT5 relative to the 1850-2018 baseline.

In addition, Figure R3 below highlights the relatively larger warming of the HadCRUT5 infilled dataset in comparison to the infilled HadCRUT4-CW dataset, when the missing values in the HadCRUT5 infilled dataset are not taken into account. Considering the uncertainties in both observed dataset, the results and conclusions of our study are not affected.

Figure R3. Trends in the Global near-Surface Air Temperature (GSAT, K decade⁻¹) calculated over the 1940-2016 period for the HadCRUT4-CW dataset (green dot-dashed line), the HadCRUT5 dataset (gray rectangle, overlying the blue dot-dashed line), the ensemble mean of the historical simulations (black dot-dashed line) and the 32 members (stars) ranked from the weakest to strongest trend. The colors of the stars display the RMSE of the GSAT (K) over the 1900-2018 period as per the color scale on the right, with respect to the HadCRUT5 dataset. A factor 1.06 is applied to observations to account for the discrepancies between GMST and GSAT metrics.

Ln 505: 'inherits' -> 'starts'

Done

Ln 522-528: This description of the regression method isn't clear to me. Some symbols seem to be missing from the PDF, for example before 'estimates' on line 527. Are both instances of xAMOC in the two equations the same thing? Overall this doesn't make sense to me. Clarify.

We apologize for the missing symbols in this method section that made it difficult to understand. Indeed, this is the same xAMOC in both equations. It corresponds to the AMOC time series over the 1940-2016 period from each of the members of the IPSL ensemble. The description of this analysis has been rewritten in the revised manuscript and is now clearer.

Reviewer #3 (Remarks to the Author):

I did not realize that this is the manuscript which I have reviewed before for Nature Climate Change. Although this manuscript is only slightly modified from the last version submitted to Nature Climate Change, to me, the findings in this manuscript are important and the manuscript is worth to be published by Nature Communication.

In this manuscript, the authors studied the potential influence of the internal variability portion of the AMOC on the simulated global mean surface temperature (GSAT) change during 1946-2018 and the implication to the relationship between observed AMOC and GSAT. They found that the weakening trend of the simulated AMOC since 1940s may have contributed to the closer to observed GSAT changes since a weaker AMOC induces a cooling effect in the Northern Hemisphere and a lessened GSAT warming. They also suggest that the ECS constrained by observations in the 20th century in previous studies may have underestimated by the possible weakening of AMOC in observations in this period. To me, the authors have done very careful analysis and comparison of the observed and model simulated data, and the conclusions reached here are also well demonstrated.

We thank the Reviewer for his/her positive evaluation.

Minor comments:

1. It is worth to explore the magnitude of the GSAT change in relation to AMOC changes in other CMIP6 models to see whether magnitude of GSAT change per 1 Sv AMOC change in IPSL-CM6A-LR model is above, below the other CMIP6 model. By doing so, one may have a sense to see whether IPSL model has significantly over or underestimated the AMOC's effect on GSAT.

This point was already explored in Fig.4a, which shows the relationship between GSAT and AMOC in the CMIP6 *piControl* simulations, with the regression coefficient ($K Sv^{-1}$) indicated beside the model's name. The IPSL-CM6A-LR model is shown in light green.

2. How the mutli-centennial AMOC variability in historical runs differ from that in the control run? Whether the members with an AMOC declining trend are affected by the AMOC initial state when these members are branched from the control run.

This is shown in Fig. R4 below. It can be seen that the AMOC in the *historical* simulations follow that from the *piControl* to some extent but some departures are also observed. Therefore, the AMOC variability can be linked to the initial state use in the historical simulations. As this analysis is interesting and fits well into the study, we added Fig. R4 in the Supplementary of the revised manuscript as Fig. S4.

Figure R4: Time evolution of the low-pass filtered AMOC (Sv) from the *piControl* simulation (black) and from the historical simulations (blue) of IPSL-CM6A-LR for the 32 members of the IPSL ensemble ranked from the first to the last member. A lanczos low-pass filter with a cutoff period of 11 years is used.

References

- Caesar, L., McCarthy, G. D., Thornalley, D. J. R., Cahill, N., & Rahmstorf, S. (2021). Current Atlantic meridional overturning circulation weakest in last millennium. *Nature Geoscience*, *14*(3), 118-120.
- Cheung, A. H., Mann, M. E., Steinman, B. A., Frankcombe, L. M., England, M. H., & Miller, S. K. (2017). Comparison of low-frequency internal climate variability in CMIP5 models and observations. *Journal of Climate*, *30*(12), 4763-4776.
- Cowan, K., & Way, R. G. (2014). Coverage bias in the HadCRUT4 temperature series and its impact on recent temperature trends. *Quarterly Journal of the Royal Meteorological Society*, *140*(683), 1935-1944.
- Fueglistaler, S. (2019). Observational evidence for two modes of coupling between sea surface temperatures, tropospheric temperature profile, and shortwave cloud radiative effect in the tropics. *Geophysical Research Letters*, *46*, 9890-9898. <https://doi.org/10.1029/2019GL083990>
- Gillett, N. P., Kirchmeier-Young, M., Ribes, A., Shiogama, H., Hegerl, G. C., Knutti, R., ... & Ziehn, T. (2021). Constraining human contributions to observed warming since the pre-industrial period. *Nature Climate Change*, *11*(3), 207-212.
- Menary, M. B., Robson, J., Allan, R. P., Booth, B. B., Cassou, C., Gastineau, G., ... & Zhang, R. (2020). Aerosol- forced AMOC changes in CMIP6 historical simulations. *Geophysical research letters*, *47*(14), e2020GL088166.
- Moberg, A., Sonechkin, D. M., Holmgren, K., Datsenko, N. M., & Karlén, W. (2005). Highly variable Northern Hemisphere temperatures reconstructed from low-and high-resolution proxy data. *Nature*, *433*(7026), 613-617.
- Peings, Y., Simpkins, G., & Magnusdottir, G. (2016). Multidecadal fluctuations of the North Atlantic Ocean and feedback on the winter climate in CMIP5 control simulations. *Journal of Geophysical Research: Atmospheres*, *121*(6), 2571-2592.
- PAGES 2k Consortium (2019) Consistent multidecadal variability in global temperature reconstructions and simulations over the Common Era. *Nature Geoscience* *12*, pp. 643–649.
- Qasmi, S., Cassou, C., & Boé, J. (2017). Teleconnection between Atlantic multidecadal variability and European temperature: Diversity and evaluation of the Coupled Model Intercomparison Project phase 5 models. *Geophysical Research Letters*, *44*(21), 11-140.
- Swingedouw D., Mignot J., Ortega P., Khodri M., Menegoz M., Cassou C. and Hanquiez V. (2017) Impact of explosive volcanic eruptions on the main climate variability modes. *Global and Planetary Changes* *150*, pp. 24-45.

Xie, S. P. (2020). Ocean warming pattern effect on global and regional climate change. *AGU advances*, 1(1), e2019AV000130.

REVIEWER COMMENTS

Reviewer #1 (Remarks to the Author):

Second review of "Increased risk of near term global warming due to a recent AMOC weakening" by Bonnet et al

The authors have addressed my concerns in the revised manuscript. The new manuscript is better referenced and the caveats are discussed better. I recommend that the manuscript be accepted for publication

Reviewer #2 (Remarks to the Author):

Review of revised version of Bonnet et al.

Overall I am happy with the authors response to my original review, but I have a number of specific comments on the manuscript and one comment on the response which I think the authors should address.

Comments on the revised manuscript:

Ln 25: "reinforces the risk" -> "means that it will be harder to avoid crossing the 2C warming threshold"

Ln 39: "a part of" -> "some of"

Ln 44: "do not allow *us* to rule out"

Ln 52: "it is interesting to focus" -> "it makes sense to focus"

Ln 58-61: I don't think this is a fully accurate description of the findings of Ribes et al. What their study actually showed was that a Mixture of two AR(1) models could not represent the internal variability well for some models, including IPSL-CM6A-LR, which is the focus of this study. And hence the study used the model fitted to observations through the rest of the study.

Ln 118: Define and explain what S_hist and TCR_hist are when first introduced.

Ln 129: Replace 'range of' with 'mean of'.

Ln 141: Writing 'a recent study' here makes it sound as though the finding that a strong AMOC causes warming in the NH is entirely new, whereas of course earlier literature has also shown this. I suggest replacing 'A recent study suggests' with 'Previous studies suggest' and add at least one additional reference on effects of the AMOC on NH climate.

Ln 147-148: 'is covering' -> 'covers'

Ln 159-160: The meaning of 'and the AMOC in global climate model simulations' here isn't entirely clear. I think what this means is that the AMOC weakening inferred from the Caesar index is consistent with weakening inferred from climate model simulations. If so, I suggest instead writing at the end of the sentence 'consistent with the trend found in global climate model simulations', and give at least one reference.

Ln 169: Replace 'S_hist or TCR_hist' with 'the calculated values of S_hist or TCR_hist'.

Ln 171-172: 'and vice versa' here means that the ensemble members with the strongest AMOC weakening are those with the weakest GSAT trends. Is this what was intended? I think what the authors probably really meant was 'and those with the strongest GSAT warming are those with the strongest AMOC strengthening'.

Ln 179: Was an objective method used to select these ensemble members? I suggest that the authors should use one, otherwise their results will depend on this subjective choice. I suggest the six ensemble members with the lowest GSAT RMSE.

Ln 191-194: This requires a reference forward to Figure 3. Also, the Caesar index, which has just been introduced, doesn't seem to show much of an increasing trend in the early 20th century in Figure 3a – can the authors comment on this?

Ln 233-235: Has this index been used before? If so, give a reference.

Ln 294: Why were only 13 of ~40 CMIP6 models chosen?

Ln 348: Delete 'the' before 'internal climate variability'.

Ln 356: Missing word after '2C warming'. Threshold?

Ln 361-364: I think it is good to cite a paper on the pattern effect here as you have done, but I would say something like 'These results are related to, but distinct from, studies which have found that the SST pattern effect in the Pacific is associated with a reduction in the observed warming trend due to internal variability'. If I understand correctly the SST pattern effect is mainly in the Pacific, where this study is focused on the Atlantic circulation. So these are distinct mechanisms, but both suggest that internal variability may have masked anthropogenic warming in recent decades.

On the response relating to Parsons et al:

Although past1000 simulations from only CMIP5 are shown by Parsons et al., their Figure 2a does allow readers to compare the CMIP6 piControl variability with paleo proxies for the 1450-1849 period. This comparison shows that some CMIP6 models overestimate variability compared to the proxies. The authors could also mention this point in their manuscript where they describe the results of Parsons et al.

In their response, the authors write 'Nevertheless, the Parsons et al. (2021) study is focusing on interdecadal variability (notably since it is only comparing model with 400-year long reconstruction, which is still a bit short to assess multi-centennial variability), while in our study, this is rather the multi-centennial variability that play a crucial role'. I don't fully understand the argument about the different timescales. The focus of the present study seems to be on understanding anomalies 1999-2018 anomalies relative to 1880-2018 (e.g. as shown in figure 1) i.e. variability on timescales less than 140 years. Parsons et al. show interdecadal variability and variability in 100-yr trends in their Figure 1, which seem like they must be quite closely related to the variability metrics used in the present study.

Reviewer #3 (Remarks to the Author):

This version of the manuscript has been significantly improved. The authors have done a very nice job to address the comments from reviewers. I would recommend this manuscript to be accepted for publication.

Response to reviews of “Increased risk of near term global warming due to a recent AMOC weakening” by Bonnet et al.

We thank the three Reviewers for their second evaluation and final suggestions that helped us to improve further the manuscript. For clarity, the Reviewers' comments are in **bold**.

Reviewer #2 (Remarks to the Author):

Overall I am happy with the authors response to my original review, but I have a number of specific comments on the manuscript and one comment on the response which I think the authors should address.

We thank the Reviewer for his positive evaluation of our work.

Comments on the revised manuscript:

Ln 25: “reinforces the risk” -> “means that it will be harder to avoid crossing the 2C warming threshold”

Done

Ln 39: “a part of” -> “some of”

Done

Ln 44: “do not allow *us* to rule out”

Done

Ln 52: “it is interesting to focus” -> “it makes sense to focus”

Done

Ln 58-61: I don't think this is a fully accurate description of the findings of Ribes et al. What their study actually showed was that a Mixture of two AR(1) models could not represent the internal variability well for some models, including IPSL-CM6A-LR, which is the focus of this study. And hence the study used the model fitted to observations through the rest of the study.

Indeed, Ribes et al. fitted the models to observations, however, the statistical model used seems to have some difficulties to reproduce the high low-frequency internal climate variability of some CMIP6 models, as stated in the introduction. So by assuming this variability is realistic and exists in the observations, their method could overconstrain future warming. Our point here is more to indicate that there is an issue with the high low-frequency internal variability of some models, including IPSL-CM6A-LR. We modified the related sentence in the revised manuscript. Please find below the new sentence.

“However, the authors showed that this method poorly accounted for the internal variability in some CMIP6 models (Ribes et al., 2021) that exhibit stronger multi-decadal to centennial internal climate variability than CMIP5 models (Parsons et al., 2020), which might have strong implications.”

Ln 118: Define and explain what S_hist and TCR_hist are when first introduced.

It is indicated line 41.

Ln 129: Replace 'range of' with 'mean of'.

Done.

Ln 141: Writing 'a recent study' here makes it sound as though the finding that a strong AMOC causes warming in the NH is entirely new, whereas of course earlier literature has also shown this. I suggest replacing 'A recent study suggests' with 'Previous studies suggest' and add at least one additional reference on effects of the AMOC on NH climate.

As this point is related specifically to the IPSL-CM6A-LR model, we do not add more references about this. We modified the beginning of the sentence, which is now: "A recent study on the associated mechanisms suggests that..." in order to better link it to the previous sentence.

Ln 147-148: 'is covering' -> 'covers'

Done.

Ln 159-160: The meaning of 'and the AMOC in global climate model simulations' here isn't entirely clear. I think what this means is that the AMOC weakening inferred from the Caesar index is consistent with weakening inferred from climate model simulations. If so, I suggest instead writing at the end of the sentence 'consistent with the trend found in global climate model simulations', and give at least one reference.

We agree that this sentence is confusing. We used "and the AMOC in global climate model simulations" to refer to the fact that the AMOC reconstruction from Caesar et al. (2018) is calibrated from the relationship between the Caesar index and the AMOC in CMIP5 climate models. We removed the "in global climate model simulations" in the revised manuscript, as our point is just to indicate on which relationship this reconstruction of the AMOC is based.

Ln 169: Replace 'S_hist or TCR_hist' with 'the calculated values of S_hist or TCR_hist'.

Done.

Ln 171-172: 'and vice versa' here means that the ensemble members with the strongest AMOC weakening are those with the weakest GSAT trends. Is this what was intended? I think what the authors probably really meant was 'and those with the strongest GSAT warming are those with the strongest AMOC strengthening'.

Indeed, our point here is that, conversely, the members with the strongest GSAT warming are those with the strongest AMOC strengthening. We clarified this point in the revised manuscript. Please find below the new sentence.

"Fig. 2a shows that ensemble members with the weakest GSAT warming are those with the strongest AMOC weakening and the members with the largest GSAT warming are those with the strongest AMOC strengthening."

Ln 179: Was an objective method used to select these ensemble members? I suggest that the authors should use one, otherwise their results will depend on this subjective choice. I suggest the six ensemble members with the lowest GSAT RMSE.

We selected these ensemble members in two steps. First we first select the members close to the observed warming trend and with a relatively strong negative AMOC trend, as suggested by the AMOC reconstructions. Then we choose the 6 members with the lowest RMSE within this subset of members. This is why the members with the 4th and the 6th lowest RMSE are not selected here, as they have a warming trend of 0.13 and 0.14 K per decade and an AMOC trend of -0.05 and -0.005 Sv per decade, which is not very consistent with the observations, with the observed warming trend of about 0.1 K per decade and AMOC trend estimated by the reconstruction of about -0.26 Sv per decade. We clarified this point in the revised manuscript. Please find below the new explanation.

“We pre-select members that are consistent with the observed GSAT and AMOC trends, and then select the six members with the lowest GSAT RMSE among those. The subset is composed of members #14, #4, #5, #25, #29 and #30, with RMSE ranked 1st, 8th, 2nd, 3rd and 5th, and 7th, respectively.”

Ln 191-194: This requires a reference forward to Figure 3. Also, the Caesar index, which has just been introduced, doesn't seem to show much of an increasing trend in the early 20th century in Figure 3a – can the authors comment on this?

In Figure 3a, the Caesar index shows an increase over the 1900 to 1940, with a trend of 0.12 K per decade in the ERSSTv5 observational dataset (Huang et al., 2017). We added a point on that in the text in order to better support this idea. Please find below the new text.

“Conversely, an internally generated enhancement of the AMOC in the early 20th century might have warmed GSAT at that time, in agreement with other studies²⁵, suggesting an important role of internal variability for this early century warming. This is consistent with the increase of the Caesar index over the beginning of the 20th century (Figure 3a).”

Ln 233-235: Has this index been used before? If so, give a reference.

Zhang (2008) and Mahajan et al. (2011) indeed highlighted the link between AMOC and the subsurface ocean temperature (at 400m) over these regions in a model, as indicated in the text line 243 of the revised manuscript, and discussed the observed trends. They used a PCA while we rather use here an index built on large scale box average. Jackson and Wood (2020) also investigate different fingerprints in more details, and confirm that large scale metrics based on temperature similar to the one used here could be used to monitor the AMOC. Even if the exact same index was not used previously, such index is quite similar to the AMOC fingerprints investigated in these previous studies.

Ln 294: Why were only 13 of ~40 CMIP6 models chosen?

As indicated in the method section, we consider CMIP6 models with Atlantic meridional stream function data available in the *piControl* simulation for at least 500 years. Thirteen models, including IPSL-CM6A-LR, were available at the time of our analysis.

Ln 348: Delete ‘the’ before ‘internal climate variability’.

Done.

Ln 356: Missing word after ‘2C warming’. Threshold?

Indeed, a word was missing, thank you for pointing out this oversight. We add “objective” instead of “threshold” in the revised version of the manuscript.

Ln 361-364: I think it is good to cite a paper on the pattern effect here as you have done, but I would say something like ‘These results are related to, but distinct from, studies which have found that the SST pattern effect in the Pacific is associated with a reduction in the observed warming trend due to internal variability’. If I understand correctly the SST pattern effect is mainly in the Pacific, where this study is focused on the Atlantic circulation. So these are distinct mechanisms, but both suggest that internal variability may have masked anthropogenic warming in recent decades.

In their study Zhou et al. (2021) do not explicitly show that the reduction in the observed warming due to the pattern effect originates from the Pacific Ocean. However, we agree that this region seems to be a large contributor to the SST pattern effect (Dong et al., 2019). We completed our paragraph with this new reference in order to better highlight that our results suggest another mechanism. Please find below the new sentences.

“These results thus seem to be in line with a recent study (Zhou et al., 2021) suggesting that the pattern effect could be in part related to internal climate variability. This internally-driven pattern effect could have masked part of the human-induced global warming in the recent decades. Nevertheless the time period investigated here is longer than the last four decades analyzed in this other study, and our mechanism is related to the North Atlantic variability, rather than the Pacific variability, as found in previous studies (Dong et al., 2019).”

On the response relating to Parsons et al:

Although past1000 simulations from only CMIP5 are shown by Parsons et al., their Figure 2a does allow readers to compare the CMIP6 piControl variability with paleo proxies for the 1450-1849 period. This comparison shows that some CMIP6 models overestimate variability compared to the proxies. The authors could also mention this point in their manuscript where they describe the results of Parsons et al.

This point was highlighted lines 377-380: “While it is usually believed that model simulations might have too low multi-centennial variability as compared to proxy records (Laepple and Huybers, 2014), a recent study (Parsons et al., 2020) suggests that GSAT interdecadal variability of some CMIP6 models might be overestimated over the period 1450-1840.”. We added “in comparison to the pre-industrial control simulations” at the end of the sentence to better describe the results of Parsons et al. (2021).

In their response, the authors write ‘Nevertheless, the Parsons et al. (2021) study is focusing on interdecadal variability (notably since it is only comparing model with 400-year long reconstruction, which is still a bit short to assess multi-centennial variability), while in our study, this is rather the multi-centennial variability that play a crucial role’. I don’t fully understand the argument about the different timescales. The focus of the present study seems to be on understanding anomalies 1999-2018 anomalies relative to 1880-2018 (e.g. as shown in figure 1) i.e. variability on timescales less than 140 years. Parsons et al. show interdecadal variability and variability in 100-yr trends in their Figure 1, which seem like they must be quite closely related to the variability metrics used in the present study.

We acknowledge that the variability studied in Parsons et al. (2021) is also relevant to our study. This is a complicated issue and we agree that we cannot exclude that the models such as IPSL-CM6A-LR have too much internal variability. We added a sentence on that point in the paragraph about the realism of the multi-centennial low-frequency internal variability of revised manuscript in order to better clarify this limitation in our study. Please find below in blue the new sentence.

“Indeed, While it is usually believed that model simulations might have too low multi-centennial variability as compared to proxy records⁴³, a recent study¹¹ suggests that GSAT interdecadal variability of some CMIP6 models might be overestimated over the period 1450-1840 in comparison to the pre-industrial control simulations. Therefore, we cannot exclude that some CMIP6 models, such as IPSL-CM6A-LR have too much internal variability.”

In our response of the previous review, we meant by the “differences of timescales” that our study overall focused on the impact of the multi-centennial variability, as we looked for long-term trends (e.g. 74 years in Figure 2a), whereas the Figure 2a of Parsons take into account variability from >25 years. Therefore, the larger interdecadal variability in the CMIP6 models in comparison to paleo proxies could be due to different timescales that the one we are interested in.

References

Caesar, L., Rahmstorf, S., Robinson, A., Feulner, G., & Saba, V. (2018). Observed fingerprint of a weakening Atlantic Ocean overturning circulation. *Nature*, 556(7700), 191-196.

Dong, Y., Proistosescu, C., Armour, K. C., & Battisti, D. S. (2019). Attributing historical and future evolution of radiative feedbacks to regional warming patterns using a Green’s function approach: The preeminence of the western Pacific. *Journal of Climate*, 32(17), 5471-5491.

Huang, B., Thorne, P. W., Banzon, V. F., Boyer, T., Chepurin, G., Lawrimore, J. H., ... & Zhang, H. M. (2017). Extended reconstructed sea surface temperature, version 5 (ERSSTv5): upgrades, validations, and intercomparisons. *Journal of Climate*, 30(20), 8179-8205.

Jackson, L. C., & Wood, R. A. (2020). Fingerprints for Early Detection of Changes in the AMOC. *Journal of Climate*, 33(16), 7027-7044.

Mahajan, S., Zhang, R., Delworth, T. L., Zhang, S., Rosati, A. J., & Chang, Y. S. (2011). Predicting Atlantic meridional overturning circulation (AMOC) variations using subsurface and surface fingerprints. *Deep Sea Research Part II: Topical Studies in Oceanography*, 58(17-18), 1895-1903.

Zhang, R. (2008). Coherent surface- subsurface fingerprint of the Atlantic meridional overturning circulation. *Geophysical Research Letters*, 35(20).

Zhou, C., Zelinka, M. D., Dessler, A. E., & Wang, M. (2021). Greater committed warming after accounting for the pattern effect. *Nature Climate Change*, 11(2), 132-136.

REVIEWERS' COMMENTS

Reviewer #2 (Remarks to the Author):

Thanks to the authors for their responses to the points raised in my second review. I think the revised manuscript is now suitable for publication.

Response to reviews of “Increased risk of near term global warming level due to a recent AMOC weakening” by Bonnet et al.

For clarity, the Reviewers' comments are in **bold**.

Reviewer #2 (Remarks to the Author):

Thanks to the authors for their responses to the points raised in my second review. I think the revised manuscript is now suitable for publication.

We thank the Reviewer for his/her positive evaluation of our work.